# Self-Healing Machine Learning: A Framework for Autonomous Adaptation in Real-World Environments

**Paulius Rauba**
University of Cambridge
pr501@cam.ac.uk

**Nabeel Seedat**
University of Cambridge
ns741@cam.ac.uk

**Krzysztof Kacprzyk**
University of Cambridge
kk751@cam.ac.uk

**Mihaela van der Schaar**
University of Cambridge
mv472@cam.ac.uk

## Abstract

Real-world machine learning systems often encounter model performance degradation due to distributional shifts in the underlying data generating process (DGP). Existing approaches to addressing shifts, such as concept drift adaptation, are limited by their *reason-agnostic* nature. By choosing from a pre-defined set of actions, such methods implicitly assume that the causes of model degradation are irrelevant to what actions should be taken, limiting their ability to select appropriate adaptations. In this paper, we propose an alternative paradigm to overcome these limitations, called *self-healing machine learning* (SHML). Contrary to previous approaches, SHML autonomously diagnoses the reason for degradation and proposes diagnosis-based corrective actions. We formalize SHML as an optimization problem over a space of adaptation actions to minimize the expected risk under the shifted DGP. We introduce a theoretical framework for self-healing systems and build an agentic self-healing solution $\mathcal{H}$-*LLM* which uses large language models to perform self-diagnosis by reasoning about the structure underlying the DGP, and self-adaptation by proposing and evaluating corrective actions. Empirically, we analyze different components of $\mathcal{H}$-LLM to understand *why* and *when* it works, demonstrating the potential of self-healing ML.

## 1 Introduction

Consider the following scenario: You are tasked with monitoring the performance of a black-box model $f$ deployed in production. After some time, you notice that the predictive performance of $f$ has started to degrade. What would be the appropriate action $a$ you should take to ensure that the model's performance returns to its prior performance levels: $a_1$: re-train the model on a subset of the data; $a_2$: change the type of the model used; $a_3$: remove discovered corrupted values; $a_4$: add new covariates?

Clearly, the answer to this question is "it depends". Different actions might result in different behavior of the model over time, as illustrated in Fig. 1. If we could pinpoint *why* the performance of the model has

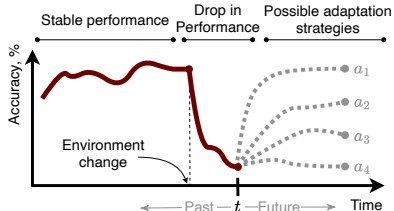

Figure 1: Different adaptation strategies $a_1, \ldots, a_4$ might result in different performance after an environment change.

degraded, it could help us understand *what* actions are most promising, since we could select an action which would directly address the root cause of the problem.

38th Conference on Neural Information Processing Systems (NeurIPS 2024).

While we take this intuition for granted, state-of-the-art techniques for handling model degradation do not reflect this line of reasoning and rely on *pre-determined actions*, such as model retraining [1–5], re-using old models [1, 6–8], or other specialized methods [6, 7, 9–12]. Such approaches share a common, implicit assumption —the *reason* for the degradation in model performance is irrelevant. We refer to this as *reason-agnostic methods*.

The practical implications of methods being *reason-agnostic* are quite concerning. By not considering the causes for drop in performance, the corrective actions are, essentially, shots in the dark. In high-stakes applications like healthcare, finance, or policing, misguided adaptations can lead to real-world harms, such as inaccurate diagnoses, financial losses, or system failures. In some industries—such as healthcare— this has resulted in *avoiding* automated model retraining altogether [13].

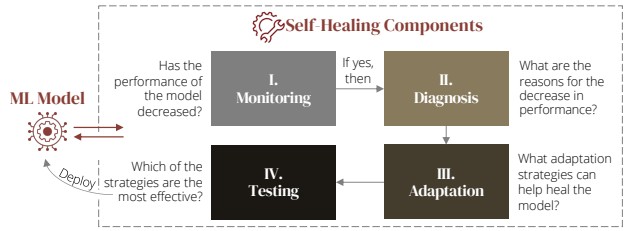

Figure 2: Our work introduces self-healing machine learning. A healing mechanism $\mathcal{H}$ interacts with a deployed model $f$. $\mathcal{H}$ contains four *components*: monitoring, diagnosis, adaptation, and testing. The overall goal of SHML is to find optimal adaptation actions to maximize the predictive performance of a model $f$.

We propose *self-healing machine learning* (SHML) to overcome the limitations of reason-agnostic approaches. SHML equips ML models with the ability to diagnose the reasons for performance degradation and take targeted corrective actions. We define a *self-healing system* as a tuple $\langle \mathcal{H}, f \rangle$, where $f$ is a black-box *model* and $\mathcal{H}$ is a *healing mechanism* that can *modulate* the behavior of $f$. An example of $\mathcal{H}$ modulating $f$ is by deciding what data to use to re-train $f$, as illustrated in our introductory example. $\mathcal{H}$ contains four *components*: *monitoring, diagnosis, adaptation*, and *testing* (Fig. 2). The goal of $\mathcal{H}$ is to decide what actions to take in response to model degradation which are chosen based on an *adaptation policy* that provides a mapping from diagnoses to actions. We therefore formalize the goal of $\mathcal{H}$ as finding optimal actions under the shifted data generating process (DGP) which are sampled from an adaptation policy conditioned on a diagnosis (Sec. 3.3). In our introductory example, the optimal action is taking action $a_1$ (Fig. 1). Building upon these insights, we propose the first self-healing ML algorithm, $\mathcal{H}$-LLM (Sec. 5) which generates diagnoses behind model degradation and suggests diagnosis-based adaptation strategies.

**Significance beyond technical contributions**. By enabling systems to autonomously diagnose and adapt to model degradation, we lay the groundwork for a new class of self-healing algorithms. We envision self-healing systems as crucial for high-stakes applications where optimal model performance is essential. We also believe this work has immediate practical relevance in high-stakes areas where model degradation is common, such as medicine [13–18], fraud detection [19] or finance [20].

**Contributions**. ① We identify fundamental limitations in existing *reason-agnostic* adaptation approaches that do not consider the reason for model degradation (Sec. 3.2). ② We introduce the paradigm of *self-healing machine learning* and establish a theoretical foundation for finding adaptation actions with diagnosis-guided action sampling (Sec. 3.3 - 4). ③ We propose the first self-healing ML algorithm $\mathcal{H}$-LLM which reasons about the causes of degradation and modulates the behavior of ML models (Sec. 5). ④ We demonstrate the viability of SHML by studying *why* and *when* it works (Sec. 6).

## 2 Related work

SHML is most closely related to *concept drift adaptation* or *specialized drift handling* methods. We provide an extended discussion on related work within each component in Appendix A.

**Concept drift adaptation.** The field of concept drift adaptation focuses on developing algorithms to maintain the performance of machine learning models in changing environments. Such algorithms are predominantly proposed within the setting of tabular data. Most common adaptation techniques are re-training models on new data [1–5, 21–24], re-using stored models [1, 6–9] or obtaining new data altogether [25, 26]. These approaches can be implicit, like continuous retraining, or explicit, based on drift detection in data or model error [5, 23, 27]. Because these approaches do not explicitly

incorporate the reason for model degradation, we refer to them as *reason agnostic*. SHML diverges from common approaches by introducing the core idea of *diagnosing the root cause* to search for optimal adaptation actions.

**Specialized drift handling.** Techniques have also been developed to adapt in the presence of various drift scenarios, such as sliding windows [10] or *adaptive classifiers* [11, 28–32] which "repair" concept drift [27]. However, these methods lack an explicit diagnosis mechanism and operate under fixed decision rules, which do not incorporate the root causes of degradation into the adaptation strategy. Similarly, works that aim to understand distribution shifts [33–35] or attribute shifts to specific variables through causal mechanisms [36, 37] provide valuable insights but do not offer a comprehensive framework for adaptation. We note that some work could be in principle a part of some SHML system components (discussed in Appendix A).

# 3 Self-healing Machine Learning

This section introduces self-healing machine learning and its four components. We present the problem setting (Sec. 3.1), explain current limitations (Sec. 3.2), and outline the four stages of SHML (Sec. 3.3). **NOTE**: Table 1 serves as a guide for navigating the paper.

| Component | Definition | Methodological contribution | Experimental contribution | Main practical implications |
|---|---|---|---|---|
| **Monitoring** | Eq. 4 | n/a | Sec. 6.3 | More robust models against false positive drift detection (Sec.6.3) |
| **Diagnosis** | Eq. 5 | Def. 1, Def. 2, Prop. 2 | Sec. 6.4 | Established framework to reason about *why* models degrade (Sec. 4) |
| **Adaptation** | Eq. 6 | Asmp. 1 | Sec. 6.5 | Targeted adaptation by identifying the root cause (Sec. 4) |
| **Testing** | Eq. 7 | Def. 3 | Sec. 6.6 | Principled framework to evaluate actions (Sec. 6.6) |
| **Self-healing ML** | Eq. 8, Sec. 3.3 | Sec. 3.3, Sec. 5.1, Sec. 5.2, Sec. 5.3 | Sec. 6.1 | New self-healing paradigm (Sec. 3.3) addressing prior limitations (Sec. 3.2); first self-healing system (Sec. 5). |

Table 1: A summary table of self-healing machine learning and its four stages, providing links to relevant sections and serving as a navigation guide for the paper.

## 3.1 Model degradation over time

**Preliminaries**. Let $\mathcal{X}$ and $\mathcal{Y}$ denote the input and output spaces, respectively, and let $\mathcal{P}_t$ denote the data distribution over $\mathcal{X} \times \mathcal{Y}$ at time step $t \in [T]$. At each $t$, we observe a batch of data $\mathcal{D}_t = \{(\mathbf{x}_t^{(i)}, y_t^{(i)})\}_{i=1}^{n_t} \sim \mathcal{P}_t^{n_t}$, where $n_t = 1$ in the streaming setting and $n_t > 1$ in the batch setting. We will drop the superscripts where clear from context.

The goal is to learn a sequence of functions $\{f_t \in \mathcal{F}\}_{t=1}^T$ that minimize the cumulative risk:

$$R(f_1, \ldots, f_T) \coloneqq \sum_{t=1}^{T} \mathbb{E}_{(\mathbf{x},y) \sim \mathcal{P}_t}[\ell(f_t(\mathbf{x}), y)] \tag{1}$$

where $\mathcal{F}$ is a function class and $\ell : \mathcal{Y} \times \mathcal{Y} \to \mathbb{R}_{\geq 0}$ is a loss function.

In the time-invariant setting where $\mathcal{P}_t = \mathcal{P} \ \forall t \in [T]$, the goal reduces to learning a single function $f^* \in \mathcal{F}$ that minimizes the risk $R(f) = \mathbb{E}_{\mathcal{P}}[\ell(f(\mathbf{x}), y)]$. When $\mathcal{P}$ is unknown, $f^*$ is often approximated by minimizing the empirical risk on a training set $\{(\mathbf{x}^{(i)}, y^{(i)})\}_{i=1}^n \sim \mathcal{P}^n$. However, when $\mathcal{P}_t$ evolves over time, the optimal predictor $f_t^* \in \arg\min_{f \in \mathcal{F}} \mathbb{E}_{\mathcal{P}_t}[\ell(f(\mathbf{x}), y)]$ changes across time steps[1]. Failing to adapt $f_t$ in this time-varying setting leads to model degradation, as the learned function becomes increasingly suboptimal w.r.t. the current data distribution.

## 3.2 Limitations of existing approaches in adapting to changing environments

Maintaining stable model performance in the presence of a changing environment poses unique challenges. As the optimal predictor $f_t^*$ evolves over time, the estimated predictor should also adapt. Ideally, we could obtain a large batch of data $\mathcal{D}_{t+1} = \{(\mathbf{x}_{t+1}^{(i)}, y_{t+1}^{(i)})\}_{i=1}^{n_{t+1}}$ and minimize the empirical risk over this dataset. However, this is often impractical due to constraints such as (i) ground-truth labels not being immediately available [38]; (ii) the streaming setting, where each new batch contains only one data point [39]; (iii) gradual shifts, where past data remains relevant [40]; or (iv) the presence of corrupted data in new batches [41].

---

[1]Three primary mechanisms through which $\mathcal{P}_t$ varies are covariate shift: $\mathcal{P}_t(\mathbf{x}) \neq \mathcal{P}_{t+1}(\mathbf{x}) \wedge \mathcal{P}_t(y|\mathbf{x}) = \mathcal{P}_{t+1}(y|\mathbf{x})$, label shift: $\mathcal{P}_t(y) \neq \mathcal{P}_{t+1}(y) \wedge \mathcal{P}_t(\mathbf{x}|y) = \mathcal{P}_{t+1}(\mathbf{x}|y)$, and concept drift: $\mathcal{P}_t(y|\mathbf{x}) \neq \mathcal{P}_{t+1}(y|\mathbf{x})$.

To address this, the research community has developed specialized methods determining the appropriate corrective actions in such drifts. As discussed in Sec. 2, these methods primarily execute pre-defined actions upon detecting a change, such as model retraining [1–5], re-using old models [1, 6–8], or other more specialized methods [6, 7, 9–12]. However, such methods are *reason-agnostic*, disregarding valuable information that inform better adaptation actions. Consider an illustrative example: suppose a batch of new data arrives, but due to a sensor malfunction [42], 80% of the labels become corrupted and are independent of the input for that batch only. Naively retraining the model on this noisy batch would degrade its performance. This is because this strategy implicitly assumes:

$$\mathbb{E}_{(\mathbf{x},y)\sim\mathcal{P}_{t+1}}[\ell(f_{t+1}(\mathbf{x}),y)] < \mathbb{E}_{(\mathbf{x},y)\sim\mathcal{P}_{t+1}}[\ell(f_t(\mathbf{x}),y)]. \tag{2}$$

where $f_{t+1}$ is a model trained on $\mathcal{D}_{t+1}$. Relying on this assumption results in *worse performance* than doing nothing. Similarly, re-using old models assumes that the past data distribution is still relevant:

$$\mathbb{E}_{(\mathbf{x},y)\sim\mathcal{P}_{t+1}}[\ell(f_{t-k}(\mathbf{x}),y)] < \mathbb{E}_{(\mathbf{x},y)\sim\mathcal{P}_{t+1}}[\ell(f_t(\mathbf{x}),y)], \tag{3}$$

for some $k > 0$. Similarly, this might result in suboptimal performance due to the nature of the shift. Each adaptation method discussed in Sec. 2 has such implicit assumptions about the model or DGP.

*By not taking into account the reason for the model degradation (such as corrupted data), the adaptation strategy is defaulting to suboptimal corrective actions.* While any adaptation strategy inherently involves some assumptions about the relationship between the predictor and the data, we would like to prioritize making *informed assumptions*. As we discuss in Sec. 3.3, a key source of such information is diagnosing *why* the model's performance has dropped.

To address the *reason-agnostic* nature of such adaptation methods, we propose a paradigm shift called *self-healing machine learning* (SHML), where deployed models autonomously diagnose the reason for degradation and take *diagnosis-guided* corrective actions.

> **Takeaway**. Existing adaptation methods make implicit, pre-defined assumptions about the nature of model degradation. Neglecting its *reason* can lead to poorly chosen actions.

### 3.3 The four stages of self-healing machine learning

Self-healing machine learning is a framework for autonomously detecting, diagnosing, and correcting performance degradation in deployed ML models. It aims to maintain model performance in changing environments without constant human intervention. The motto of self-healing ML is "understanding your problem is half the solution" (and the most important half). A SHML system is defined by a tuple $\langle \mathcal{H}, f \rangle$, where $f : \mathcal{X} \to \mathcal{Y}$ is the deployed machine learning model we aim to heal (i.e., the function that makes predictions on input data), and $\mathcal{H}$ is a healing mechanism that interacts with the environment and acts upon the model $f$ by proposing and implementing actions, such as selecting when to retrain a model, what data to use or how to change the input data before making predictions. Thus, $\mathcal{H}$ can *modulate* the behavior of the deployed model $f$.

> **Self-Healing Machine Learning in a nutshell.**
>
> Self-healing ML contains four components: monitoring, diagnosis, adaptation, and testing. After these steps, the best action is implemented on the ML model, illustrated in Fig. 2.
>
> **I. Monitoring**. The first step is the detection of degradation, potentially due to a shift in the data distribution. We formalize this as a *monitoring* component $\mathcal{H}_M$ that takes as input the sequence of data batches $\{\mathcal{D}_i\}_{i=1}^t$, up to time $t$, and outputs $s_t \in [0,1]$, indicating the likelihood of model degradation. Formally,
>
> $$\mathcal{H}_M : (\mathcal{X} \times \mathcal{Y})^* \to [0,1], \tag{4}$$
>
> where higher values of $s_t$ indicate a greater likelihood of a shift.
>
> **II. Diagnosis**. The diagnosis component $\mathcal{H}_D$ detects the reason of degradation. It takes data batches $\{\mathcal{D}_i\}_{i=1}^t$, up to time $t$, along with any available contextual information $c \in \mathcal{C}$ (e.g. background knowledge), and outputs a distribution $\zeta \in \Delta(\mathcal{Z})$ over a space of possible reasons $\mathcal{Z}$:
>
> $$\mathcal{H}_D : (\mathcal{X} \times \mathcal{Y} \times \mathcal{C})^* \to \Delta(\mathcal{Z}). \tag{5}$$
>
> $\mathcal{Z}$ represents the finite space of possible reasons of the shift and $\zeta$ is a stochastic vector.
>
> **III. Adaptation**. The adaptation component is a policy $\pi$ that outputs a distribution over actions. Given a diagnosis vector $\zeta$, actions $a \in \mathcal{A}$ are selected from a finite space $\mathcal{A}$ by:

$$a \sim \pi(\cdot|\zeta), \text{ where } \pi : \Delta(\mathcal{Z}) \to \Delta(\mathcal{A}). \tag{6}$$

Each action $a$ modifies $f$. We denote the model used at time $t$, selected by action $a$, as $f_a^t$.

**IV. Testing**. The *testing* component $\mathcal{H}_T$ evaluates each action $a \in \mathcal{A}$ on a relevant distribution and outputs a performance measure:

$$\mathcal{H}_T : \mathcal{A} \times \mathcal{P} \to \mathbb{R}. \tag{7}$$

**Objective**. The goal of self-healing ML is to select the optimal action $a^*$ that minimizes the expected loss $\mathbb{E}[\ell(f_t^a(\mathbf{x}), y)]$ on the data distribution $\mathcal{P}_t$:

$$a^* = \arg\min_{a \in \mathcal{A}} \mathbb{E}_{(\mathbf{x},y) \sim \mathcal{P}_t}[\ell(f_t^a(\mathbf{x}), y)], \tag{8}$$

where $\ell : \mathcal{Y} \times \mathcal{Y} \to \mathbb{R}_{\geq 0}$ is a loss function, and $f_t^a$ denotes the model selected by action $a$ to be used at time $t$. The action $a$ is selected according to the adaptation policy $\pi(\cdot|\zeta)$, which maps the diagnosis vector $\zeta$ (a distribution over possible reasons for degradation) to a distribution over actions.

Suppose the following motivating example to guide the notation above.

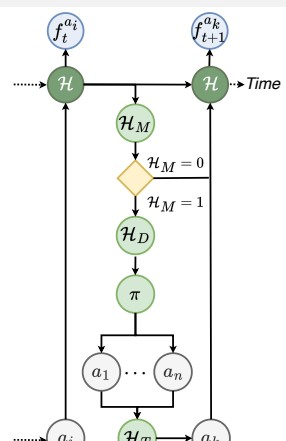

**Illustrative example**. Consider a deployed ML model $f_t$ for predicting diabetes. The monitoring component $\mathcal{H}_M$ detects a significant drop in performance, with $s_t = 0.95$ (Eq. 4). The diagnosis component $\mathcal{H}_D$ outputs three most likely reasons $z_1, z_2, z_3 \in \mathcal{Z}$, with $\zeta(z_1) = 0.95$ for data quality issues, $\zeta(z_2) = 0.03$ for concept drift, and $\zeta(z_3) = 0.02$ for model overfitting (Eq. 5). Based on, $\zeta$, the adaptation policy $\pi$ samples two actions $a_1, a_2 \sim \pi(\cdot|\zeta)$ (Eq. 6): $a_1$: remove detected biologically implausible values (e.g. Age > 200) and retrain $f_t$; $a_2$: include interaction terms between features to capture non-linearities. The testing component $\mathcal{H}_T$ evaluates the adapted models $f_t^{a_1}$ and $f_t^{a_2}$ on new incoming data (Eq. 7) and selects $a_1$ due to lower estimated loss.

The primary insight of SHML is that the action $a \sim \pi(\cdot|\zeta)$ should be based on the diagnosis $\zeta$, which is a distribution over possible reasons $z \in \mathcal{Z}$ for model degradation. In contrast, standard approaches (Sec. 3.2) assume that $\pi \perp\!\!\!\perp \zeta$. SHML formalizes this as an optimization problem over a space of adaptation actions—we aim to find the optimal actions to take each time the model $f$ degrades, with these actions chosen by the policy $\pi$ of the self-healing system $\mathcal{H}$ (Fig. 3). Different policies $\pi_1$ and $\pi_2$ might propose different actions in response to the same performance drop. While the diagnosis $\zeta$ informs the policy, we do not assume it is necessarily useful. Since $\zeta \in \Delta(\mathcal{Z})$ is a probability distribution, it can encode no knowledge by being uniform over the diagnosis space: $\zeta(z) = \frac{1}{|\mathcal{Z}|}, \forall z \in \mathcal{Z}$. These components and their interactions between two time points $t$ and $t+1$ are shown in Fig. 3.

Figure 3: The self-healing mechanism $\mathcal{H}$ modulates the function $f$ via four stages. The chosen adaptation action $a$ is implemented onto the function $f$ at the next time step.

The effectiveness of the adaptation actions depends on the diagnosis, i.e. how well can we identify the root cause. Therefore, we turn to the *diagnosis* component next.

**Takeaway**. SHML is a framework which selects actions based on the reason for model degradation. It contains four stages: monitoring, diagnosis, adaptation, and testing.

## 4 An analysis of the properties of self-healing diagnosis

Self-healing systems have the unique property of having a diagnosis stage. But what constitutes a good diagnosis? In this section, we analyze the properties of self-healing diagnosis and establish its connection to the performance of adaptation actions.

To effectively use diagnosis information to guide the search for adaptation actions, we require a way to quantify the usefulness of a diagnosis. We propose three desirable properties for such a measure: (i) **concentration**: it should favor diagnoses that provide more information, i.e. assign higher probabilities to fewer possible reasons; (ii) **sensitivity**: it should be sensitive to changes in the diagnosis distribution, such that small changes in probabilities would result in small changes in the measure; (iii) **maximum uncertainty**: it should reach its maximum value when the diagnosis

distribution is uniform, indicating no knowledge about the reason for degradation. Therefore, we propose using the entropy of the diagnosis vector as a useful proxy for quality which satisfies all three properties. Because entropy measures uncertainty, we refer to this as the *certainty of the diagnosis*.

**Definition 1** (Certainty of the Diagnosis). *Let $\mathcal{Z}$ be the finite space of possible reasons for degradation and $\Delta(\mathcal{Z})$ be the diagnosis space. The certainty of a diagnosis $\zeta \in \Delta(\mathcal{Z})$ in a self-healing machine learning system is measured by its entropy $\mathbb{H}(\zeta)$, defined as:*

$$\mathbb{H}(\zeta) = -\sum_{z \in \mathcal{Z}} \zeta(z) \log \zeta(z), \tag{9}$$

*where $\zeta(z)$ is the probability of reason $z$ under the distribution $\zeta$.*

The link between diagnosis quality and adaptation performance highlights the importance of obtaining informative diagnoses in SHML. Building upon these concepts, we define the optimal diagnosis:

**Definition 2** (Optimal Diagnosis). *The optimal diagnosis $\zeta^*$ is defined as:*

$$\zeta^* = \arg\min_{\zeta \in \Delta(\mathcal{Z})} \mathbb{E}_{a \sim \pi(\cdot|\zeta)}[R(a)] \tag{10}$$

*where $\Delta(\mathcal{Z})$ is the diagnosis space, $\pi(\cdot|\zeta)$ is the conditional distribution over actions induced by diagnosis vector $\zeta$, and $R(a)$ denotes the risk of $f_t^a$ associated with action $a \in \mathcal{A}$.*

This formalizes the intuition that the best diagnosis is the one that leads to the best adaptation actions, on average. To characterize the properties of this optimal diagnosis, we introduce an assumption about the structure of the adaptation policy:

**Assumption 1** (Independent actions). *We assume that $\pi(\cdot|\zeta)$ has a hierarchical structure. First, a reason $z \in \mathcal{Z}$ is sampled according to the diagnosis $\zeta$: $z \sim \zeta$. Then, an action is sampled conditioned on this reason, $a \sim \pi(\cdot|z^\dagger)$, where $z^\dagger \in \Delta(\mathcal{Z})$ such that $z^\dagger(z) = 1$. $\pi(\cdot|\zeta)$ can then be described as the following mixture.*

$$\pi(a|\zeta) = \sum_{z \in \mathcal{Z}} \pi(a|z^\dagger)\zeta(z) \tag{11}$$

*$\zeta$ are the mixture weights and $\{\pi(\cdot|z^\dagger) \mid z \in \mathcal{Z}\}$ are the mixture components.*

Under this hierarchical structure, we can prove a useful property of the optimal diagnosis:

**Proposition 1.** *Under Assumption 1, the optimal diagnosis $\zeta^*$ has a zero entropy, i.e., $\mathbb{H}(\zeta^*) = 0$.*

Proof in Appendix E. To ensure that the optimal diagnosis is well-defined, we also prove its existence under mild assumptions:

**Proposition 2** (Existence of Optimal Diagnosis). *Suppose that the action space $\mathcal{A}$ is a compact subspace of $\mathbb{R}^n$ and $R$ is continuous. Then there exists at least one optimal diagnosis $\zeta^*$.*

*Proof.* The expected risk $\mathbb{E}_{a \sim \pi(\cdot|\zeta)}[R(a)]$ is a continuous function of $\zeta$ by the continuity of $R$ and the compactness of $\mathcal{A}$. Since the diagnosis space $\Delta(\mathcal{Z})$ is also compact (being a probability simplex), the extreme value theorem guarantees the existence of a minimizer $\zeta^*$. $\qquad\square$

**Takeaway**. The existence of an optimal diagnosis establishes a foundation for designing algorithms that can accurately approximate it in practice. By identifying the underlying reasons for performance degradation, a high-quality diagnosis enables a self-healing system to take the most effective adaptation actions.

## 5    Building self-healing systems: $\mathcal{H}$-LLM

This section outlines the challenges of building SHML systems (Sec. 5.1), describes how LLMs can address these challenges (Sec. 5.2), and introduces the first self-healing system, $\mathcal{H}$-LLM (Sec. 5.3).

### 5.1    Unique challenges of building self-healing systems

Implementing SHML systems (Sec. 3.3) poses unique challenges in diagnosis and adaptation.

**Challenges in diagnosis**. Discovering the reasons for model degradation poses significant practical challenges because: (i) the space of possible reasons $\mathcal{Z}$ is often poorly defined or intractable to

| Diagnosis | Evidence | Confidence (1-10) |
|---|---|---|
| There are outliers in the data | The maximum values for many variables in the new dataset are significantly higher than in the old dataset, suggesting the presence of outliers | 9 |
| Incorrect data transformations have been applied | The mean values for many variables in the new dataset are significantly higher than in the old dataset, suggesting that the data may have been incorrectly transformed | 8 |
| There are data entry errors | The minimum value for Insulin in the new dataset is negative, which is not possible in a real-world context | 10 |

Table 2: Example diagnoses suggested by $\mathcal{H}$-LLM. The system proposes diagnoses and suggests evidence for the diagnosis. A post-hoc relative confidence score, constructed using the "evidence" column, helps to guide which diagnoses to pay most attention to while designing adaptation policies.

specify exhaustively in real-world scenarios; and (ii) assigning well-calibrated probabilities to reasons for model degradation is difficult due to both the epistemic and aleatoric uncertainty that exists in real-world environments. This makes it difficult to approximate the optimal diagnosis (Def. 2).

**Challenges in adaptation**. The adaptation policy $\pi$ (Eq. 6) requires selecting optimal adaptation actions $a$ based on the diagnosis $\zeta$. This is challenging because (i) it requires reasoning about how actions interact with diagnoses; and (ii) the space of adaptation actions may be extremely large in practice, making it difficult to find the optimal action (Eq. 8).

## 5.2 Language models to empower self-healing

We posit that LLMs have the potential to satisfy many of the required properties of self-healing components because of the following capabilities: (i) **Hypothesis proposers**. LLMs are known to be "phenomenal hypotheses proposers" [43] which are required to hypothesizing diagnoses of ML model performance degradation; (ii) **Contextual understanding**. LLMs have been pretrained with a vast corpus of information and hence have extensive prior knowledge around different contexts and settings [44, 45]; (iii) **Language model agents**. Language models can work as agents within a larger system [46, 47] which is required to actively interact with a deployed model, trigger and implement changes. We therefore see LLMs as capable proxies for different self-healing components.

## 5.3 Design of $\mathcal{H}$-LLM

We instantiate the healing mechanism $\mathcal{H}$ with an LLM $l$, using its useful properties (Sec. 5.2) to address the practical challenges of designing SHML systems (Sec. 5.1).

---

**$\mathcal{H}$-LLM in a nutshell.**

$\mathcal{H}$-LLM is the first SHML algorithm that modulates the behavior of $f$ following Fig. 3.

**I. Monitoring**. We use statistical drift detection algorithms to monitor model degradation from $k$ previous time points [29, 39, 48]. Diagnosis is triggered if a shift is detected.

**II. Diagnosis**. Upon detection, we use a pre-defined prompt template to obtain information about the dataset before and after the diagnosis. The prompt template gives us numerical insights into how the dataset has changed and includes covariate information before and after the shift, together with other numerical details. We denote this prompt as an extractor function $\mathcal{E} : \mathcal{D}^* \to \mathcal{D}_c$ to obtain an information vector $\mathbf{v}$. Using $\mathbf{v}$ and a chain-of-thought (CoT) module with self-reflection, $\mathcal{H}$-LLM generates $k$ candidate reasons for degradation $\{\mathbf{z}_i\}_{i=1}^{k} \sim l(\cdot|\mathbf{v})$ via Monte Carlo (MC) sampling with associated confidence scores. As before, this is obtained by following pre-defined prompt templates *conditioned on* the obtained information (e.g. "*Suggest {self.n} possible reasons why the model might have failed on the basis of the issues presented*"). These candidates form an empirical diagnosis vector $\hat{\zeta}$, approximating the optimal diagnosis $\zeta^*$. Table 2 illustrates diagnoses generated by $\mathcal{H}$-LLM.

**III. Adaptation**. Conditioned on the empirical diagnosis distribution $\hat{\zeta}$, $\mathcal{H}$-LLM generates $m$ candidate adaptation actions $\{a_j\}_{j=1}^{m} \sim l(\cdot|\hat{\zeta})$ via CoT-based MC sampling. This approximates sampling from $\pi(\cdot|\zeta^*)$ (Def. 6). The actions sampled from $l$ are textual representations, so we use an interpreter function to execute each $a$ on $f$.

**IV. Testing**. The sampled actions are evaluated on an empirical dataset (Def. 7), and the empirically optimal action $\hat{a}^* = \arg\min_{j \in [m]} R(a)$ is implemented. Limited access to the shifted DGP complicates evaluating $R(a)$, but it can be approximated with empirical data $\hat{\mathcal{D}}_{\text{test}}$ by using *a backtesting window*, *continuously incoming data*, or *historical data* (Appendix B.4).

**Goal**. This procedure aims to approximate the optimal action (Def. 8). These actions are orchestrated by an orchestrator component in $\mathcal{H}$-LLM which can navigate between these steps.

---

Appendix B provides an extended discussion of $\mathcal{H}$-LLM, including the algorithm, prompts, examples, and outputs. The following table links $\mathcal{H}$-LLM with the theoretical framework.

| Component | Theory | $\mathcal{H}$-LLM | Approximation |
|---|---|---|---|
| Monitoring | Monitor for degradation | Drift detection algorithm | Any detection algorithm |
| Diagnosis | Optimal diagnosis $\zeta^* \in \Delta(\mathcal{Z})$ | Empirical diagnosis via LLM $\hat{\zeta}$ | MC sampling with LLM |
| Adaptation | Sample action $a \sim \pi(\cdot\|\zeta)$ | Sample actions via LLM $a \sim l(\cdot\|\hat{\zeta})$ | MC sampling with LLM |
| Testing | Evaluate each $a$ on $\mathcal{P}_t$ | Evaluate each $a$ on $\hat{\mathcal{D}}_{\text{test}}$ | Any suitable dataset |

**Table 3:** A comparison of the theoretical components, their implementation, and their approximations.

# 6 Experimental viability studies

The previous sections constituted the primary contribution of our paper—establishing SHML as a framework. The goal of this section is to provide a *viability* study by analyzing different components of SHML. We conduct six viability studies.[2]

**Experimental setup**. We desire to meet two properties: (i) have full control of the DGP to vary experimental parameters; and (ii) we need to benchmark against existing adaptation methods (Sec. 2) which are predominantly tabular-based. Therefore, we simulate a diabetes prediction task [49–51] based on the setup in Sec. 3.1. We predict diabetes $Y_t \in \{0,1\}$ at each time point $t$ for a set of $n$ observations, generated according to a (changing) pre-specified DGP $\log\left(\frac{P(Y_t=1|X_t)}{P(Y_t=0|X_t)}\right) = \alpha_t + \sum_{k\in K} \beta_{t,k} X_{t,k} + \epsilon_t$, where $K$ includes relevant parameters such as Age or BMI, $\beta_{t,k}$ are time-varying covariates and $\epsilon_t \sim \mathcal{N}(0, \sigma^2)$ is a noise component. For evaluating $\mathcal{H}$-LLM actions, we use a *backtesting window*—a representative sample of the shifted distribution obtained after detecting the change but before deploying the adapted model (Sec. B.4). Details provided in Appendix C.

## 6.1 Viability study I: Adaptation in the presence of model degradation

**Setup**. We aim to empirically demonstrate the limitations of existing approaches in adapting to changing environments (Sec. 3.2). We benchmark $\mathcal{H}$-LLM against four common drift adaptation methods: (i) *new model retraining* on post-drift data, (ii) *partially updating* models with new data; (iii) *Ensemble methods* by re-using old models, and (iv) *No retraining* of the models [1]. At time $t$, we introduce a sudden, single intervention by changing the DGP parameters *and* corrupting a percentage $\tau$ of $k$ columns. Table 4 shows the performance of different methods across $\tau$ and $k$.

| | **Number of corrupted columns $k$ ($\tau = 0.05$)** | | | | **Corruption percentage $\tau$ ($k = 3$)** | | | | |
| *Method* | 2 | 4 | 6 | 8 | 0.01 | 0.02 | 0.05 | 0.10 | 0.20 |
|---|---|---|---|---|---|---|---|---|---|
| No retraining | $0.44 \pm 0.02$ | $0.44 \pm 0.02$ | $0.45 \pm 0.02$ | $0.45 \pm 0.02$ | $0.43 \pm 0.02$ | $0.44 \pm 0.02$ | $0.44 \pm 0.02$ | $0.45 \pm 0.02$ | $0.46 \pm 0.02$ |
| Partially Updating | $0.71 \pm 0.02$ | $0.69 \pm 0.02$ | $0.67 \pm 0.02$ | $0.54 \pm 0.06$ | $0.74 \pm 0.03$ | $0.72 \pm 0.02$ | $0.70 \pm 0.02$ | $0.66 \pm 0.02$ | $0.62 \pm 0.02$ |
| New model training | $0.70 \pm 0.02$ | $0.69 \pm 0.02$ | $0.67 \pm 0.02$ | $0.50 \pm 0.02$ | $0.77 \pm 0.02$ | $0.74 \pm 0.02$ | $0.69 \pm 0.02$ | $0.66 \pm 0.02$ | $0.61 \pm 0.02$ |
| Ensemble Method | $0.70 \pm 0.02$ | $0.69 \pm 0.02$ | $0.67 \pm 0.02$ | $0.50 \pm 0.02$ | $0.77 \pm 0.02$ | $0.74 \pm 0.02$ | $0.69 \pm 0.02$ | $0.66 \pm 0.02$ | $0.61 \pm 0.02$ |
| $\mathcal{H}$-**LLM** | $\mathbf{0.93 \pm 0.01}$ | $\mathbf{0.87 \pm 0.01}$ | $\mathbf{0.79 \pm 0.02}$ | $\mathbf{0.68 \pm 0.02}$ | $\mathbf{0.95 \pm 0.01}$ | $\mathbf{0.94 \pm 0.01}$ | $\mathbf{0.90 \pm 0.02}$ | $\mathbf{0.82 \pm 0.02}$ | $\mathbf{0.70 \pm 0.02}$ |

Table 4: Accuracy of a deployed model $f$ upon an intervention which changes the DGP and corrupts $\tau$ percentage of $k$ columns. Error represents standard deviation. $\uparrow$ is better.

**Discussion**. The performance of $f$ degrades if the corrupted columns are not handled appropriately, such as removing or inputting the corrupted data. Defaulting to standard techniques of adapting to a changed environment results in poor performance. $\mathcal{H}$-LLM diagnoses issues by observing that some values have drifted too much from their original values *and* the DGP has changed. One of the proposed adaptation strategies is to remove samples which were estimated to be corrupted, and re-training the model on the remainder of the data. This results in superior performance.

**Takeaway 1**. Diagnosing the root cause of degradation can guide better adaptation actions.

## 6.2 Viability study II: Adaptation across datasets

**Setup**. We aim to empirically analyze whether SHML can provide benefits across different datasets. We cover five different datasets: Airlines [52], Poker [53], Weather [54], Electricity [55], Forest Type

---

[2]Code can be found at: `https://github.com/pauliusrauba/Self_Healing_ML` or `https://github.com/vanderschaarlab/Self_Healing_ML`

[56]. We simulate real-world unexpected degradations by assuming lagged labels and corrupting features at test time and evaluating models for different datasets (Table 5).

**Discussion**. Ground-truth labels are often not immediately available [38], a core feature of many streaming settings. We evaluate how $\mathcal{H}$-LLM compares to existing approaches in such scenarios. Across five datasets with different characteristics, $\mathcal{H}$-LLM consistently outperforms traditional adaptation methods by adapting $f_t^a$ at each time point $t$. Therefore, SHML's ability to identify and decorrupt features provides a robust adaptation strategy across varied data distributions and schemas.

**Takeaway 2**. Identifying the root cause and restoring features dacn provide consistent benefits across datasets.

| Method | Accuracy when $k = 5$ | | | | | Accuracy when $\tau = 5$ | | | | |
|---|---|---|---|---|---|---|---|---|---|---|
| | airlines | poker | weather | elec | covType | airlines | poker | weather | elec | covType |
| No retraining | $0.53 \pm 0.01$ | $0.48 \pm 0.01$ | $0.57 \pm 0.05$ | $0.66 \pm 0.04$ | $0.51 \pm 0.03$ | $0.53 \pm 0.01$ | $0.47 \pm 0.01$ | $0.59 \pm 0.04$ | $0.67 \pm 0.03$ | $0.58 \pm 0.01$ |
| Partially Updating | $0.53 \pm 0.01$ | $0.48 \pm 0.01$ | $0.57 \pm 0.05$ | $0.66 \pm 0.04$ | $0.51 \pm 0.03$ | $0.53 \pm 0.01$ | $0.47 \pm 0.01$ | $0.59 \pm 0.04$ | $0.67 \pm 0.03$ | $0.58 \pm 0.01$ |
| New model training | $0.54 \pm 0.02$ | $0.49 \pm 0.01$ | $0.56 \pm 0.03$ | $0.66 \pm 0.05$ | $0.51 \pm 0.02$ | $0.53 \pm 0.02$ | $0.47 \pm 0.00$ | $0.60 \pm 0.02$ | $0.67 \pm 0.03$ | $0.58 \pm 0.02$ |
| Ensemble Method | $0.51 \pm 0.01$ | $0.48 \pm 0.01$ | $0.58 \pm 0.06$ | $0.57 \pm 0.09$ | $0.52 \pm 0.02$ | $0.52 \pm 0.01$ | $0.46 \pm 0.00$ | $0.59 \pm 0.05$ | $0.65 \pm 0.04$ | $0.59 \pm 0.01$ |
| $\mathcal{H}$-**LLM** | $\mathbf{0.56 \pm 0.00}$ | $\mathbf{0.70 \pm 0.03}$ | $\mathbf{0.66 \pm 0.02}$ | $\mathbf{0.72 \pm 0.01}$ | $\mathbf{0.73 \pm 0.00}$ | $\mathbf{0.56 \pm 0.00}$ | $\mathbf{0.70 \pm 0.03}$ | $\mathbf{0.66 \pm 0.02}$ | $\mathbf{0.72 \pm 0.01}$ | $\mathbf{0.73 \pm 0.00}$ |

Table 5: Accuracy of various methods on different datasets with corrupted columns and varying corruption values. Error represents standard deviations of five runs. ↑ is better. We simulate concept drift by corrupting the test set as follows: we randomly selected $k$ features and multiplied their values by a corruption factor $\tau$. $\mathcal{H}$-LLM identifies that the test data has been corrupted and perform a relevant transformation to *decorrupt* the value to its original feature space at test time.

## 6.3 Viability study III: Monitoring

**Setup**. We use the same setup as experiment I and vary the drift detection threshold which influences the sensitivity of a detection system to changes in the DGP. Low values mean high sensitivity, and high values mean low sensitivity [57]. We measure *average recovery time* for $\mathcal{H}$-LLM to return recover from degradation and *post-intervention accuracy*, $\mathcal{H}$-LLMs average performance after intervention. Fig. 4 shows this relationship.

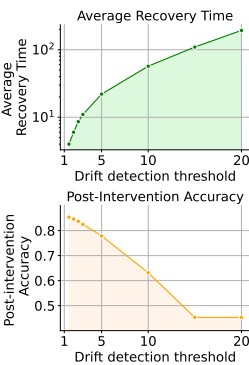

**Discussion**. Intuitively, one might expect earlier drift detection (lower threshold) to consistently yield faster recovery and higher accuracy. In reality, concept drift algorithms often struggle with false positives which can result in *worse* model performance because of unnecessary re-training [5, 58]. Self-healing ML exhibits greater robustness to these false positives, as any action will be implemented only if it outperforms doing nothing. This contrasts with traditional systems which would automatically trigger the selected action. In Fig. 4, this represents the higher post-intervention accuracy with smaller thresholds.

Figure 4: Lower drift detection thresholds can benefit SHML.

**Takeaway 3**. SHML has greater robustness to implementing poor adaptation actions.

## 6.4 Viability study IV: Diagnosis

**Setup**. We evaluate how well self-healing systems identify the root causes of problems. We corrupt a proportion of observations (*corruption coefficient*) by multiplying their values by a factor (*outlier factor*) and see if the $\mathcal{H}$-LLM detects issues related to these factors. We output a probability distribution over diagnoses of which variable is corrupted. Knowing the true corrupted variable, we measure the difference between the distributions using KL-Divergence, with lower values indicating closer matches to true corruption. A uniform diagnosis baseline represents random guessing. Fig. 5 shows these differences.

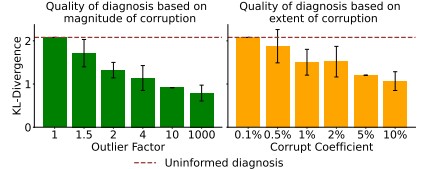

Figure 5: KL-Divergence between estimated probabilities of which variables are corrupted, and true probabilities, based on *outlier factors* and *corruption coefficients*. ↓ is better.

**Discussion**. As the *outlier factor* and *corruption coefficient* increase, making data issues more apparent, $\mathcal{H}$-LLM assigns higher probabilities to the corrupted variables. Thus, the diagnosis accuracy improves as the problem becomes more evident.

**Takeaway 4**. The quality of the diagnosis improves when the issues become more apparent.

## 6.5 Viability study V: Adaptation

**Setup**. We study the sensitivity of SHML adaptation actions by examining how well actions perform based on (i) the number of corrupted values and (ii) the size of the backtesting dataset. Fig. 6 shows this relationship.

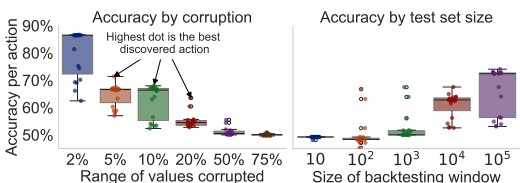

**Discussion**. As more values are corrupted, adaptation actions become more concentrated and less effective. With a larger backtesting dataset, actions are more spread out. This suggests (i) action evaluation is more reliable with non-corrupted data and (ii) larger backtesting windows help in selecting better adaptations.

Figure 6: Model $f$ accuracy for actions with varying corruption range and backtesting window size.

**Takeaway 5**. A large test dataset and high-quality data can improve adaptation action selection.

## 6.6 Viability study VI: Testing

**Setup**. We study the importance of the testing component (Eq. 7) by evaluating $\mathcal{H}$-LLM suggested actions with and without the testing phase (backtesting window) and comparing their accuracies. Fig. 7 shows this relationship.

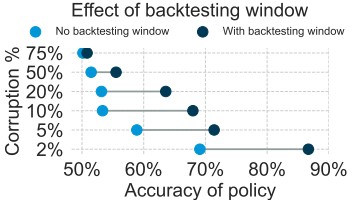

**Discussion**. Having a dataset to evaluate actions significantly improves the self-healing process. We see that better data quality results in more reliable adaptation policies.

**Takeaway 6**. The testing component is important to effectively evaluate the proposed actions.

Figure 7: Accuracies of optimal actions with and without testing components

## 6.7 Other studies

We provide further experiments in Appendix D. Our framework shows strong performance with lower warm-start parameters and increasing benefit as data degradation becomes more severe (Sec. D.4). A component-wise ablation analysis (Sec. D.6) reveals each stage of SHML is essential. Extended benchmarks (Sec. D.5) and model agnostic evaluations (Sec. D.7) demonstrate consistent improvements across different adaptation approaches and ML architectures.

## 7 Discussion

Algorithms hold significant decision-making power in high-stakes applications, yet little has been done to ensure their optimal performance. This work presents a major leap towards that goal. By enabling systems to autonomously diagnose and adapt to new environments, we aim to create a wave of self-healing systems beneficial to both the ML community and society. Our theoretical framework (Sec. 3.3, 4) builds the foundation for the development of self-healing theory, such as optimal adaptation or diagnosis methods, and our viability study shows the potential benefits of SHML. Our largest contribution is formalizing this field—we hope to spur new theoretical developments and encourage the adoption of such systems in critical domains like medicine [13–18] and finance [20].

**Limitations**. SHML's success relies on accurate root cause identification and finding effective adaptation policies which could pose challenges in some complex, real-world settings (Sec. 5.1). Furthermore, the prioritization of adaptation strategies is also not trivial. Currently, $\mathcal{H}$-LLM primarily looks for subgroup-level issues. We see future work tackling all areas of self-healing ML: finding better diagnosis strategies, improving adaptation selection, and enabling better testing of actions in the presence of changing environments.

**Broader impact**. Because SHML could empower many *positive* technologies, it could also be *misused* to amplify the impact of more problematic systems, such as surveillance technologies.

## Acknowledgements

We would like to thank the anonymous reviewers, Julianna Piskorz, Katarzyna Kobalczyk, Haris Mackevicius, and Andrew Rashbass for their helpful feedback. PR is supported by GSK, KK is supported by Roche, NS by the Cystic Fibrosis Trust. This work was supported by Microsoft's Accelerate Foundation Models Academic Research initiative.

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

# Appendix: Self-Healing Machine Learning: A framework for autonomous adaptation in real-world environments

## Table of Contents

# A Extended related work

In this section, we describe and contrast our work with other related areas.

## A.1 Comparison to other fields

**Concept drift adaptation.** Concept drift adaptation algorithms, a key component of self- healing ML systems, primarily handle drifts by re-training models on new data [1–5] or older, pre-trained stored models [1, 6–8]. These approaches can be implicit, like continuous retraining, or explicit, based on drift detection in data or model error [5, 23, 27]. Drift detection methods compare distributions, analyze data sequentially, or use statistical process control [59]. For instance, the DDM algorithm [23] has in-control, warning, and out-of-control states.

**Specialized drift handling.** Techniques have been developed for various drift scenarios. For recurring drifts, methods store and reuse historical models [6, 7, 9]. Streaming data is handled by blind approaches, like sliding windows [10] or adaptive decision trees [11], and informed approaches with explicit drift detection [21–24]. Resampling can repair adaptation errors [27], while dynamic classifier selection finds the best model for each input [60]. Methods have been proposed for robustness to noise [12], specific drift types [61], and other issues [62, 63]. Recent work explores understanding distribution shifts through latent variable models [34] and other techniques [35]. Some adaptive methods of re-training the model also include adding more hidden layer to a learner upon detection of a drift [62, 63]. Another area of research closely linked within the field is dynamic selection which attempts to find the most suitable classifier conditional on the covariates [60].

**On "repairing concept drift".** There have been other methods propose that implicitly try to adapt by detecting changes [27]. However, these adaptations are still based on the observed empirical distributions as opposed to observing the reason for degradation. By periodically sampling the accuracy of inactive classifers, the authors identify cases where change was missed or misclassifed. However, this falls under the broader umbrella of trying out many pre-determined actions without directly reasoning about the reason for model degradation.

**Continual learning.** One might get the impression that self-healing machine learning might bear close resemblance to continual learning. Continual learning focuses on developing models that learn continuously from a stream of data, acquiring, retaining, and transferring knowledge across tasks over time [64]. This contrasts strongly with self-healing machine learning. Below, **we outline seven criteria by which self-healing machine learning and continual learning differ**.

---

**Differences between continual learning and self-healing machine learning**.

1. **Objective**. The objectives of the two fields are different. Continual learning aims to learn sequentially from a stream of tasks while mitigating catastrophic forgetting. SHML focuses on autonomously diagnosing and recovering from performance degradation within a single task due to distribution shifts.

2. **Knowledge retention**. A core goal of continual learning is to preserve previously acquired knowledge while learning new tasks. SHML does not explicitly aim to retain prior knowledge or acquire new knowledge, but rather to maintain stable performance on the current task by adapting to the reason for degradation.

3. **Stability-Plasticity Dilemma**. Continual learning grapples with the trade-off between being plastic enough to learn new tasks and stable enough to remember old ones. In contrast, there is no such dilemma within SHML.

4. **Task expansion**. Continual learning seeks to expand the model's capabilities by increasing the number of tasks it can perform. In contrast, SHML operates on a single well-defined task—ensuring the optimal performance of a model, typically by minimizing empirical risk— and does not aim to increase the number of tasks. Instead, the focus is on ensuring optimal performance under a single task.

5. **Adaptation mechanism**. The underlying logic or mechanism of adaptation is different. Continual learning typically adapts by modifying model architecture, updating parameters via constrained optimization, using memory replay. In contrast, SHML explicitly adapts by diagnosing the root cause of performance drops and conditioning an adaptation action on the basis of that diagnosis. This explicit mechanism which is conditioned on is not a part of a continual learning system.

---

6. **Shift assumptions**. Continual learning primarily handles shifts across distinct tasks, where the input or output distribution changes between tasks. In contrast, SHML considers shifts within the same task, where the joint distribution might change.

7. **Theoretical formalism**. Continual learning is often formalized as a sequence of constrained optimization problems to mitigate interference between tasks. In contrast, SHML is formalized as finding an optimal policy that can propose actions on the basis of diagnoses.

## A.2   Comparison on a component level

Here, we focus on some related work within each sub-component. Table 6 provides key related work within each column. We do not focus separately on *adaptation* and *testing* because adaptation is covered above, whereas testing is simply a stage which helps to evaluate proposed actions.

| component | Definition | Methodological contribution | Experimental contribution | Main practical implications | Related work |
|---|---|---|---|---|---|
| Monitoring | Eq. 4 | n/a | Sec. 6.3 | More robust models against false positive drift detection (Sec.6.3) | [4, 23, 24, 29, 39, 48] |
| Diagnosis | Eq. 5 | Def. 1, Def. 2, Prop. 2 | Sec. 6.4 | Established framework to reason about *why* models degrade (Sec. 4) | [36, 37] |
| Adaptation | Eq. 6 | Asmp. 1 | Sec. 6.5 | Targeted adaptation by identifying the root cause (Sec. 4) | [1, 21, 27, 30–32, 61] |
| Testing | Eq. 7 | Def. 3 | Sec. 6.6 | Principled framework to evaluate actions (Sec. 6.6) | [65–67] |
| Self-healing ML | Eq. 8, Sec. 3.3 | Fig. 1, Fig. 3, Sec. 3.3, Sec. 5.1, Sec. 5.2, Sec. 5.3 | Sec. 6.1 | New self-healing paradigm (Sec. 3.3) addressing prior limitations (Sec. 3.2); first self-healing system (Sec. 5). | [68–72] |

Table 6: Summary table of self-healing ML. Use this as a guiding source to navigate the paper. Related work defines the most similar available work within each component

**Monitoring**. Related work within monitoring largely relates to different statistical techniques for discovering the presence of shifts/drifts or model degradation. We see them as an integral part of SHML. However, they are also actively used by other adaptation methods to trigger adaptation systems.

**Diagnosis**. The diagnosis component is a core component of SHML. Two primary works are closely related. The first work, "why did the distribution change?" [36], attempts to factorize the change of the joint distribution into conditional distributions of each variable and attribute some changes to one of the marginals. This is achieved by modeling the change and relationship between variables as a causal mechanism. The second work, "why did the model fail?" [37] attributes model performance degradation via a causal mechanism. They assume that distribution shifts are induced due to an intervention in the causal mechanism which results in model performance changes, and uses Shapley values to attribute changes to specific distributions. These two methods are fundamentally different from SHML in multiple respects. First and most important, these works do not propose any actions on the basis of these failures or shifts. The primary goal of both works is to understand why a distribution has changed or a model has failed, attributing it to a causal mechanism, instead of *adapting* the model to perform optimally. Second, the theoretical formalism introduced is substantially different and comes with different properties. Both works operate within the directed acyclic graph (DAG) framework, whereas we operate under a diagnosis component which is defined as a vector over a space of possible reasons. Other key differences relate to the adaptation mechanism, shift assumptions, adaptation assumptions, level of granularity of the diagnosis, level of granularity of the adaptation, or testing.

Recent work has already started coming out on *understanding* distribution shifts [35]. It is known that understanding why a distribution shift happens is important for mitigating that shift [35]. Some other people have looked at modeling shifts via latent variable models without relying on access to labels at test time [34]. However, as before, these methods do not share the objective of finding optimal actions for adaptation.

**Self-healing systems outside ML**. Self-healing systems have been proposed outside of machine learning [68–72]. We view these as inspirations for our work but consider them disparate and separate because none of them touch upon the core problem of machine learning model degradation, and have not been applied in practice.

### A.3 Unique properties of self-healing machine learning

The core of self-healing machine learning revolves around two primary components: the deployed ML model $f$ and the healing system $\mathcal{H}$. Here, we provide additional clarity on these components and their interactions:

**Definition of the Deployed Model $f$** The model $f$ represents the deployed machine learning model that we aim to heal. It is the function that makes predictions on input data and whose performance we're trying to maintain and improve. In our viability studies, we demonstrate this framework using logistic regression models as $f$, though the approach generalizes to any predictive model.

**Relationship Between $f$ and $\pi$** While $f$ is the model making predictions, $\pi$ is the adaptation policy—a function that determines what actions to take to modify $f$ based on the diagnosed reasons for its performance degradation. The healing system $\mathcal{H}$ follows policy $\pi$ to output actions $a$ (such as $a_1$: retrain a model or $a_2$: remove corrupted features) which are then implemented onto $f$. Therefore, $\mathcal{H}$ follows policy $\pi$ which helps to determine optimal actions $a$ that change/modulate the deployed ML model $f$.

**Practical Implementation** In our viability studies with H-LLM, the policy $\pi$ is instantiated with an LLM (GPT-4) which uses the diagnosed reasons for model failures (also achieved with an LLM) to propose concrete actions. For instance, if $f$ is a diabetes prediction model and $\pi$ diagnoses that $f$'s performance has degraded due to concept drift, $\pi$ might suggest an action to retrain $f$ with more recent data or to adjust feature weights.

# B $\mathcal{H}$-LLM

This section provides more details on $\mathcal{H}$-LLM.

## B.1 Algorithm and details H-LLM

The algorithm of $\mathcal{H}$-LLM is presented in Algorithm 1.

**Extended discussion**.

**I. Monitoring**. We use statistical drift detection algorithms to monitor model degradation from $k$ previous time points [29, 39, 48]. Diagnosis is triggered if a shift is detected. For our practical implementation, we use the Drift Detection Method, a popular method for binary drift detection classification.

**II. Diagnosis**. Upon detection, $\mathcal{H}$-LLM uses an extractor function $\mathcal{E} : \mathcal{D}^* \to \mathcal{D}_c$ to transform the dataset information into an information vector $\mathbf{v}$. This extractor function is a mapping from the dataset to information about the dataset. It takes information before the shift happened and calculates summary statistics, such as the mean, average, standard deviation, percentiles, etc., within each column, as well as the performance of a deployed model $f$ under various data slices. For instance, this would also involve looping over all variables, binning them into 10 discrete values and calculating the average model performance across each bin. This is done to ensure that the information contained within the information vector are both summary statistics, i.e. how the

---

**Algorithm 1: $\mathcal{H}$-LLM**

**Require :** $f, \mathcal{H}_M, \mathcal{H}_D, \mathcal{H}_A, \mathcal{H}_T, \tau, m, k$

$t \leftarrow 1$
$t^* \leftarrow$ null
**while** $t \leq T$ **do**
    $s_t \leftarrow \mathcal{H}_M(\{(\mathbf{x}_i, y_i)\}_{i=1}^t)$
    **if** $s_t > \tau$ *and* $t^*$ = *null* **then**
        $t^* \leftarrow t$
    **if** $t^* \neq$ *null and* $t - t^* >$ *Detection Window*
      **then**
        $t' \leftarrow t$
        $\mathbf{v} \leftarrow \mathcal{E}((\mathbf{x}, y) \sim \mathcal{P}_{t^*}, c \in \mathcal{C})$
        **for** $i = 1$ *to* $k$ **do**
            $\mathbf{z}_i \sim l(\cdot|\mathbf{v})$
        $\hat{\zeta} \leftarrow \{\mathbf{z}_i\}_{i=1}^k$
        **for** $j = 1$ *to* $m$ **do**
            $a_j \sim l(\cdot|\hat{\zeta})$
        $\hat{a}^* \leftarrow \arg\min_{j \in [m]} \mathcal{H}_T(f^{a_j}, \hat{\mathcal{D}}_{[t^*, t']})$
        $f \leftarrow f^{\hat{a}^*}$
        $t^* \leftarrow$ null
    $t \leftarrow t + 1$

---

data has changed, as well as specific performance metrics within data slices. The information is used to generate specific diagnoses as to what has happened. We observe, for instance, that summary statistics are extremely helpful if there are any larger deviations from average, as the diagnosis module within $\mathcal{H}$-LLM picks up on these clues. This information is provided as textual information to the next step which is the diagnosis phase.

The information vector is used as a textual representation within the next LLM call to generate concrete hypotheses / diagnoses about the reason for the $f$ failure. This is where additional context $c$ could be added, if available, such as the presence of any particular exogenous events that could have affected model performance and could guide the diagnosis search. In the future, we envision that the additional context could be acquired by the system itself. This is used in a chain-of-thought module with self-reflection, where $k$ candidates for degradation are generated along with associated scores. We employ different "diagnosis" modules within $\mathcal{H}$-LLM. For instance, there is a specific diagnosis module that only attempts to find which covariates are responsible for degradation. The system level instruction could be as follows: "Find covariates that are responsible for the model degrading". However, we also supplement this with more broader reasons for degradation, such as "Find and hypothesize reasons that could have resulted in model degradation , given the information provided". We provide three prompt templates used to hypothesize issues in Section B.2.1. We sample such prompts $m$ times using MC sampling. The chain-of-thought and self-reflection is implemented by calling $\mathcal{H}$-LLM multiple times to re-consider the evidence and hypotheses. Table 2 illustrates diagnoses generated by $\mathcal{H}$-LLM.

**III. Adaptation**. Conditioned on the empirical diagnosis distribution $\hat{\zeta}$, $\mathcal{H}$-LLM generates $m$ candidate adaptation actions $\{a_j\}_{j=1}^m \sim l(\cdot|\hat{\zeta})$ via CoT-based MC sampling. Specifically, we focus on three kinds of adaptation actions.

- Generic adaptation actions

- Adaptation actions by removing corrupted data
- Adaptation actions by training multiple models for subsets of the data

This is reflected in three different prompt templates in Appendix B.2.2.

**Generic adaptation actions**. The first attempt is to find generic adaptation actions that the diagnosis module suggests on the basis of the identified evidence. These are often quite generic, for instance, "add new covariates that could control for the seasonality". In many such cases, within the confines of our experiments, we do not have the ability to resolve the issues on the basis of the proposed solutions. Therefore, we add two more directly actionable adaptation actions that are also attempted by $\mathcal{H}$-LLM *after* the generic adaptation actions have been attempted.

**Adaptation actions by removing corrupted data**. Another concrete adaptation action is that we instruct $\mathcal{H}$-LLM to hypothesize specific data slices that might have been corrupted. This could be, for instance, biologically implausible values (negative insulin, age > 200, implausible hba1c levels), mismatches (e.g. height, weight do not match BMI), sudden shifts in the data (ages change from averages of 30 to 60), and other. The adaptation module then proposes *which data slices to remove* to achieve superior performance. These suggested data slices are then removed and re-trained in the next batch.

**Adaptation actions by training multiple models**. The final concrete adaptation action is to propose specific data slices where the model might have drifted within that slice. This is done because instead of *global drifts*, models sometimes drift locally and require complete re-training of the new dataset.

Example outputs of such strategies are presented in Appendix B.3. We note, however, that, in reality, there might be many possible adaptation actions, such as re-training the model on combinations of old and historical data, re-using old models, re-using parts of old models, creating custom ensembles, changing models altogether, changing hyperparameters or adding regularization terms, building different models for different samples based on their difficulty, switching between symbolic and predictive ML models in the face of high uncertainty, and many more. Our approach is to introduce only the primary few ways with the hope of extending this in the future.

As before, because the actions sampled from $l$ are textual representations, we use an interpreter function to execute each $a$ on $f$.

**IV. Testing**. The sampled actions are evaluated on an empirical dataset (Def. 7), and the empirically optimal action $\hat{a}^* = \arg\min_{j\in[m]} R(a)$ is implemented. Limited access to the shifted DGP complicates evaluating $R(a)$, but it can be approximated with empirical data $\hat{\mathcal{D}}_{\text{test}}$ by using *a backtesting window*, *continuously incoming data*, or *historical data*. In all of our experiments, we use a backtesting window. However, other strategies could be attempted. The different strategies are explained in greater detail in Appendix B.4.

**Goal**. This procedure aims to approximate the optimal action (Def. 8). We remark that there might be better adaptation policies that could be suggested on the basis of evidence. Likewise, there might be better diagnosis modules available. We see $\mathcal{H}$-LLM as a first attempt to integrate self-healing into ML.

## B.2   Prompt templates used

The following are some of the primary prompt templates used within $\mathcal{H}$-LLM.

### B.2.1   Prompts related to diagnosis

```
"""
        Given the following information:
        - Data before the shift: {x_before.describe()}
        - Data after the shift: {x_after.describe()}
        - Context: {context}
        - Model performance across each covariate before the shift: {
    covariate_performance_before}
        - Model performance across each covariate after the shift: {
    covariate_performance_after}

```

```
 9          You know for a fact that the model has degraded. Analyze the
        covariates and think why.
10
11          Review each existing covariate and provide a hypothesis on
        whether it might have changed and resulted in the model
        underperforming. Provide evidence for each hypothesis and the
        strength of belief for each covariate.
12
13          Format your output as follows:
14          Covariate: <covariate>; Hypothesis: ...; Evidence: ...;
        Strength of belief: ...
15
16          After reviewing all the covariates, assign a confidence score
        for each covariate indicating your confidence level that the
        covariate has issues. Use the following confidence levels:
        extremely confident, confident, somewhat confident, unsure,
        completely unsure. Only use 'extremely confident' if you have
        overwhelming evidence for your decision. Prioritize making more
        confident beliefs. Avoid being uncertain. Use the available inputs
         as well as the data to make the best possible decision. Your goal
         is to be correct while reducing entropy of the probabilities (be
        confidently correct).
17          """
18 """
```

Code Listing 1: Generic diagnosis prompt

```
 1 """
 2          Given the following information:
 3          - Data before the shift: {x_before.describe()}
 4          - Data after the shift: {x_after.describe()}
 5          - Context: {context}
 6          - Model performance across each covariate before the shift: {
        covariate_performance_before}
 7          - Model performance across each covariate after the shift: {
        covariate_performance_after}
 8
 9          You know for a fact that the model has degraded. Analyze the
        covariates and think why.
10
11          Then, hypothesize {n} possible covariates or combinations of
        covariates that might have changed and resulted in the model
        underperforming. Each possibility should be mutually exclusive.
        For example, [X1] is one possibility, [X2] is another, and [X1, X2
        ] is a third.
12          """
```

Code Listing 2: Generic diagnosis prompt for searching combinations of covariates responsible for degradation

```
 1 """
 2          Given the following information:
 3          - Data before the shift: {x_before.describe()}
 4          - Data after the shift: {x_after.describe()}
 5          - Context: {context}
 6          - Initial hypotheses on covariates or combinations of
        covariates that might have changed and resulted in model
        underperformance: {covariate_guesses}
 7
 8          Summarize the provided hypotheses and assign probabilities to
        each hypothesis such that the total probability sums to 100%.
 9
10          Your probabilities should be reflective of the evidence and
        data. Uniform probabilities (10% each) implies no knowledge. 100%
        probability on one covariate implies certain belief. Prioritize
```

```
       making more confident beliefs. Avoid being uncertain. Use the
       available inputs as well as the data to make the best possible
       decision. Your goal is to be correct while reducing entropy of the
        probabilities (be confidently correct).
11
12         Format each hypothesis and its probability as follows:
13         Hypothesis: [<covariate1>, <covariate2>, ...]; Probability: <
       probability>
14         """
```

Code Listing 3: Diagnosis probability prompt

### B.2.2 Prompt templates related to adaptation

```
1          """
2
3         Suppose the following hypothesized issues in the dataset: {
       issues}
4          Data before the shift: {x_before.describe()}
5          Data after the shift: {x_after.describe()}
6
7          Suggest {self.n} possible reasons why the model might have
       failed on the basis of the issues presented. These reasons should
       be hypotheses that might have resulted in the degradation of the
       model if such hypotheses turn out to be true. These hypotheses
       also have to be likely on the basis of the issues provided. These
       hypotheses should be specific to the data itself. The goal is to
       track down specific changes within the data that could have
       resulted in the model degradation.
8
9          Format your output as follows:
10
11         Hypothesis: <>; Evidence: <>
12
13         """
```

Code Listing 4: Generic adaptation prompt

```
1
2          f"""
3          Suppose the following issues in the dataset: {issues}
4          Data before the shift: {x_before.describe()}
5          Data after the shift: {x_after.describe()}
6
7          Suggest {self.n} possible subgroups that if removed could
       result in better performance for the model.
8          The subgroups can be single (e.g. X > x) but could also be
       multiple combinations (e.g. X > x and Y < y)
9          """
```

Code Listing 5: Subgroup adaptation prompt

```
1          """Suggest solutions based on removing data"""
2          task = f"""
3          Suppose the following issues in the dataset: {issues}
4          Data before the shift: {x_before.describe()}
5          Data after the shift: {x_after.describe()}
6
7          Suggest {self.n} possible subgroups that might need re-
       training. That is, fitting a separate model on these subgroups
       might result in superior performance.
8          The subgroups can be single (e.g. X > x) but could also be
       multiple combinations (e.g. X > x and Y < y)
9          """
```

Code Listing 6: Subgroup retrain prompt

## B.3 Example outputs of H-LLM

```
1  """
2  Covariate: HbA1c; Hypothesis: The distribution of HbA1c has shifted
       after the shift, which might have resulted in the model
       underperforming; Evidence: The mean of HbA1c has slightly
       increased from 5.699 to 5.730, and the standard deviation has also
        increased from 0.492 to 0.505. The model performance across
       different ranges of HbA1c has significantly dropped after the
       shift; Strength of belief: Extremely Confident
3
4  Covariate: FastingGlucose; Hypothesis: The distribution of
       FastingGlucose has shifted after the shift, which might have
       resulted in the model underperforming; Evidence: The mean of
       FastingGlucose has slightly increased from 99.639 to 100.519, and
       the standard deviation has slightly decreased from 15.341 to
       15.280. The model performance across different ranges of
       FastingGlucose has significantly dropped after the shift; Strength
        of belief: Extremely Confident
5
6  Covariate: Age; Hypothesis: The distribution of Age has shifted after
       the shift, which might have resulted in the model underperforming;
        Evidence: The mean of Age has slightly increased from 49.721 to
       49.900, and the standard deviation has slightly increased from
       12.066 to 12.310. The model performance across different ranges of
        Age has significantly dropped after the shift; Strength of belief
       : Extremely Confident
7
8  Covariate: BMI; Hypothesis: The distribution of BMI has shifted after
       the shift, which might have resulted in the model underperforming;
        Evidence: The mean of BMI has slightly decreased from 25.075 to
       24.985, and the standard deviation has slightly decreased from
       4.015 to 3.903. The model performance across different ranges of
       BMI has significantly dropped after the shift; Strength of belief:
        Extremely Confident
9
10 Covariate: BloodPressure; Hypothesis: The distribution of
       BloodPressure has shifted after the shift, which might have
       resulted in the model underperforming; Evidence: The mean of
       BloodPressure has slightly increased from 119.417 to 120.106, and
       the standard deviation has increased from 14.177 to 15.205. The
       model performance across different ranges of BloodPressure has
       significantly dropped after the shift; Strength of belief:
       Extremely Confident
11
12 Covariate: Cholesterol; Hypothesis: The distribution of Cholesterol
       has shifted after the shift, which might have resulted in the
       model underperforming; Evidence: The mean of Cholesterol has
       slightly decreased from 201.108 to 198.927, and the standard
       deviation has slightly increased from 39.557 to 40.269. The model
       performance across different ranges of Cholesterol has
       significantly dropped after the shift; Strength of belief:
       Extremely Confident
13
14 Covariate: Insulin; Hypothesis: The distribution of Insulin has
       shifted after the shift, which might have resulted in the model
       underperforming; Evidence: The mean of Insulin has slightly
       increased from 84.667 to 84.925, and the standard deviation has
       slightly decreased from 45.090 to 45.057. The model performance
       across different ranges of Insulin has significantly dropped after
        the shift; Strength of belief: Extremely Confident
15
16 Covariate: PhysicalActivity; Hypothesis: The distribution of
       PhysicalActivity has shifted after the shift, which might have
       resulted in the model underperforming; Evidence: The mean of
```

```
            PhysicalActivity has slightly decreased from 3.080 to 3.020, and
            the standard deviation has slightly increased from 0.994 to 1.014.
             The model performance across different ranges of PhysicalActivity
             has significantly dropped after the shift; Strength of belief:
            Extremely Confident.
17  """
18
19  #### Generated probability outputs based on the above
20
21  # Prompt for getting these hypotheses:
22
23  """
24  Your goal is to provide a summary of probabilities on likelihood of
            each of the covariates resulting in the model failing
25  """
26
27  Outputs:
28
29  Hypothesis: [HbA1c]; Probability: 12.5%
30  Hypothesis: [FastingGlucose]; Probability: 12.5%
31  Hypothesis: [Age]; Probability: 12.5%
32  Hypothesis: [BMI]; Probability: 12.5%
33  Hypothesis: [BloodPressure]; Probability: 12.5%
34  Hypothesis: [Cholesterol]; Probability: 12.5%
35  Hypothesis: [Insulin]; Probability: 12.5%
36  Hypothesis: [PhysicalActivity]; Probability: 12.5%
```

Code Listing 7: Output for guesses which covariates have shifted. This example showcases that when there is little evidence that any specific covariate has shifted more than the others

```
1  """
2  1. Issue: Increase in standard deviation; Evidence: The standard
        deviation for most of the variables has increased in the new
        dataset, indicating increased variability in the data; Confidence:
         8
3  2. Issue: Change in mean values; Evidence: The mean values for most of
         the variables have changed, which could indicate a shift in the
        population being studied; Confidence: 7
4  3. Issue: Change in minimum and maximum values; Evidence: The minimum
        and maximum values for most of the variables have changed, which
        could indicate outliers or a change in the range of data;
        Confidence: 7
5  4. Issue: Change in quartile values; Evidence: The 25%, 50%, and 75%
        quartile values for most of the variables have changed, indicating
         a change in the distribution of the data; Confidence: 7
6  5. Issue: Negative values for Insulin and PhysicalActivity; Evidence:
        The minimum values for Insulin and PhysicalActivity are negative,
        which is not possible in a real-world scenario and indicates data
        errors; Confidence: 10
7  6. Issue: Change in distribution of data; Evidence: The changes in
        mean, standard deviation, and quartile values suggest a change in
        the distribution of the data, which could affect the model's
        performance; Confidence: 8
8  7. Issue: Increase in age range; Evidence: The minimum and maximum age
         has increased, indicating a broader age range in the new dataset;
         Confidence: 6
9  8. Issue: Decrease in BMI; Evidence: The mean BMI has decreased in the
         new dataset, which could indicate a change in the health status
        of the population being studied; Confidence: 6
10 9. Issue: Increase in Blood Pressure; Evidence: The mean Blood
        Pressure has increased in the new dataset, which could indicate a
        change in the health status of the population being studied;
        Confidence: 6
```

```
11  10. Issue: Decrease in Cholesterol; Evidence: The mean Cholesterol has
        decreased in the new dataset, which could indicate a change in
        the health status of the population being studied; Confidence: 6
12  """
```

Code Listing 8: Generic issue response which identifies overall issues within the dataset.

```
1   """
2   1. Subgroup: Individuals with age > 85; Reason: The maximum age has
       increased in the new dataset, which could be due to outliers or
       errors in the data.
3   2. Subgroup: Individuals with age < 12; Reason: The minimum age has
       decreased in the new dataset, which could be due to outliers or
       errors in the data.
4   3. Subgroup: Individuals with Insulin < 0; Reason: Negative values for
        Insulin are not possible in a real-world scenario and indicate
       data errors.
5   4. Subgroup: Individuals with PhysicalActivity < 0; Reason: Negative
       values for PhysicalActivity are not possible in a real-world
       scenario and indicate data errors.
6   5. Subgroup: Individuals with BMI < 12.8; Reason: The minimum BMI has
       decreased in the new dataset, which could be due to outliers or
       errors in the data.
7   6. Subgroup: Individuals with BloodPressure < 70.5; Reason: The
       minimum Blood Pressure has decreased in the new dataset, which
       could be due to outliers or errors in the data.
8   7. Subgroup: Individuals with Cholesterol < 66.3; Reason: The minimum
       Cholesterol has decreased in the new dataset, which could be due
       to outliers or errors in the data.
9   8. Subgroup: Individuals with FastingGlucose > 154; Reason: The
       maximum FastingGlucose has increased in the new dataset, which
       could be due to outliers or errors in the data.
10  9. Subgroup: Individuals with HbA1c < 4; Reason: The minimum HbA1c has
        decreased in the new dataset, which could be due to outliers or
       errors in the data.
11  10. Subgroup: Individuals with BMI > 39.6; Reason: The maximum BMI has
        increased in the new dataset, which could be due to outliers or
       errors in the data.
12  """
```

Code Listing 9: Example response about which subgroups to remove

```
1   """
2   1. Subgroup: Individuals with age > 85; Reason: The maximum age has
       increased in the new dataset, indicating a broader age range.
3   2. Subgroup: Individuals with age < 12; Reason: The minimum age has
       decreased in the new dataset, indicating a broader age range.
4   3. Subgroup: Individuals with BMI < 12.83; Reason: The minimum BMI has
        decreased in the new dataset, indicating a change in the health
       status of the population.
5   4. Subgroup: Individuals with BMI > 37.07; Reason: The maximum BMI has
        increased in the new dataset, indicating a change in the health
       status of the population.
6   5. Subgroup: Individuals with Blood Pressure > 166.85; Reason: The
       maximum Blood Pressure has increased in the new dataset,
       indicating a change in the health status of the population.
7   6. Subgroup: Individuals with Blood Pressure < 70.49; Reason: The
       minimum Blood Pressure has decreased in the new dataset,
       indicating a change in the health status of the population.
8   7. Subgroup: Individuals with Cholesterol < 44.64; Reason: The minimum
        Cholesterol has decreased in the new dataset, indicating a change
        in the health status of the population.
9   8. Subgroup: Individuals with Cholesterol > 347.08; Reason: The
       maximum Cholesterol has increased in the new dataset, indicating a
        change in the health status of the population.
```

```
10  9. Subgroup: Individuals with Insulin < -79.81; Reason: The minimum
       Insulin has decreased in the new dataset , indicating a data error.
11  10. Subgroup: Individuals with PhysicalActivity < -0.30; Reason: The
       minimum PhysicalActivity has decreased in the new dataset ,
       indicating a data error.
12  """
```
Code Listing 10: Example response about which subgroups to retrain the model on

## B.4   Evaluation strategies of self-healing algorithms

Self-healing relies on a testing phase, i.e. the ability to test whether the proposed actions perform well on a test dataset. However, given that the distribution has shifted and the historical data no longer represents the new distribution, one might ask: how can we test models on this new distribution? The primary alternative used in our experiments is a backtesting window which we define formally below.

**Definition 3** (Backtesting Window). *Let $\{\mathcal{P}_t\}_{t \in \mathbb{T}}$ be a sequence of probability measures on $\mathcal{X} \times \mathcal{Y}$, and suppose a distributional shift occurs at time $t^* \in \mathbb{T}$, i.e., $\mathcal{P}_{t^*} \neq \mathcal{P}_{t^*-1}$. Let $t' > t^*$ be the time at which the self-healing system detects the shift. The **backtesting window** is the time interval $[t^*, t']$ satisfying the following properties:*

$$\forall t \in [t^*, t'] : (\mathbf{x}_t, y_t) \sim \mathcal{P}_{t^*},$$
$$\forall t \in [t^*, t'] : (\mathbf{x}_t, y_t) \nsim \mathcal{P}_{t^*-1}.$$

We notice that the backtesting window is a unique property that arises uppon sudden shifts in the data generating process. Specifically, because we assume only two data generating processes and a transition between them at time point $t$, then all points $k$ where $k > t$ will be from the new DGP and all points $k < t$ will be from the old DGP. Since a drift detection algorithm requries some time to detect the drift, by the time a drift has been detected, we have some collected data from the new distribution which we call the *backtesting window*. We can therefore optimize our actions on this specific window of the dataset.

Clearly, this does not hold when the assumptions about the nature of the shift change. In such a case, we could always use continuously incoming streaming data. Upon the arrival of each new batch, we can test each proposed action and validate it, consistently upgrading and using the actions that perform well on the most recent batch of data. This strategy assumes that the labels are almost immediately available at prediction time. If not, another strategy employed could be to test such actions on the mot recent available data with labels.

Other approaches could include generating synthetic data to imitate the new shift with labels or using historical data by de-biasing it. However, these are experimental approaches which need further validation.

## B.5   Computational notes

**Computational overhead.** SHML methods have larger overhead than reason-agnostic approaches due to the self-healing system (LLM pipeline) identifying model failure reasons. Practically, it takes 20-40 seconds to implement a full pipeline and correct a model upon drift detection. This overhead is negligible for real-world systems given the benefits. Overhead may vary across systems.

**Sample efficiency.** No differences exist as failure detection doesn't depend on sample size, but on self-healing pipeline complexity.

# C  Case study design

Code can be found at: `https://github.com/pauliusrauba/Self_Healing_ML` or `https://github.com/vanderschaarlab/Self_Healing_ML`

## C.1  Details on the experimental setup

**Experimental setup**. To evaluate the performance of self-healing systems, we require to manipulate the data generating process (DGP) and ask "what-if" questions. Real-world datasets, while valuable, do not offer control over the DGP and come with pre-embedded biases that can implicitly affect detection systems [73]. In contrast, by using synthetic data to control the DGP, we can run controlled *in silico* experiments and perform viability studies [74]. Furthermore, the overwhelming majority of model adaptation methods are designed for tabular data (refer to Sec. 2 and Sec. A) which includes our benchmarks (see Sec. 6.1). Therefore, we simulate a diabetes prediction task [49–51]. We perfectly mimic the introduced setup in Sec. 3.1. Our goal is to predict the presence of diabetes $Y_t \in \{0, 1\}$ at each time point $t$ for a set of $n$ observations, generated according to a (changing) pre-specified DGP $\log \left( \frac{P(Y_t=1|X_t)}{P(Y_t=0|X_t)} \right) = \alpha_t + \sum_{k \in K} \beta_{t,k} X_{t,k} + \epsilon_t$, where $K$ includes relevant covariates such as Age or BMI, $\beta_{t,k}$ are time-varying covariates and $\epsilon_t \sim \mathcal{N}(0, \sigma^2)$ is a noise component.

We generated synthetic data for the diabetes prediction task. Each feature is sampled from a normal distribution with specified parameters:

- Hemoglobin A1c (HbA1c) levels are sampled from a normal distribution: HbA1c $\sim \mathcal{N}(5.7, 0.5^2)$.
- Fasting Glucose levels are sampled from a normal distribution: Fasting Glucose $\sim \mathcal{N}(100, 15^2)$.
- Age is sampled from a normal distribution: Age $\sim \mathcal{N}(50, 12^2)$.
- Body Mass Index (BMI) is sampled from a normal distribution: BMI $\sim \mathcal{N}(25, 4^2)$.
- Blood Pressure is sampled from a normal distribution: Blood Pressure $\sim \mathcal{N}(120, 15^2)$.
- Cholesterol levels are sampled from a normal distribution: Cholesterol $\sim \mathcal{N}(200, 40^2)$.
- Insulin levels are sampled from a normal distribution: Insulin $\sim \mathcal{N}(85, 45^2)$.
- Physical Activity is sampled from a normal distribution: Physical Activity $\sim \mathcal{N}(3, 1^2)$.

The observations $X$ are constructed as a matrix where each row is an instance of the generated features. The outcomes are then determined by running the model through a logistic regression and obtaining a binary outcome value.

## C.2  Details on viability studies

### C.2.1  Viability Study I

**Viability Study I**. To simulate covariate shift and introduce data corruption, we follow these steps:

1. Generate two datasets with different coefficients and noise parameters:
   - The first dataset with $n_1 = 100,000$ samples, coefficients $\beta_1 = [0.3, 0.0075, -0.01, 0.05, 0.04, -0.03, -0.02, -0.1]$, and noise $\epsilon_1 \sim \mathcal{N}(0, 0.2^2)$.
   - The second dataset with $n_2 = 100,000$ samples, coefficients $\beta_2 = [-0.3, -0.0075, 0.2, -0.05, -0.015, -0.001, 0.02, -2]$, and noise $\epsilon_2 \sim \mathcal{N}(0, 0.2^2)$.
2. Split the first dataset into training and testing sets, using a 70/30 split.
3. Combine the testing set of the first dataset with the entire second dataset to form the complete testing set. The second testing set therefore contains a shift where the transitions between the DGPs happen.
4. In addition to the shift in the DGP, we introduce outliers in the second dataset by multiplying selected features by an outlier factor that control. By default, it is set to

5. This outlier factor corrupts the number of columns corrupted by $k$, and corrupts a percentage of values within the column, denoted as $\tau$.

6. The shift index is determined as the starting point of the second dataset in the combined testing set.

7. We measure and report the performance of the model during the second data generating process.

**Summary of parameters**:

- $n_1 = 100,000$: Number of samples in the first dataset.
- $n_2 = 100,000$: Number of samples in the second dataset.
- $\beta_1 = [0.3, 0.0075, -0.01, 0.05, 0.04, -0.03, -0.02, -0.1]$: Coefficients for the first dataset.
- $\beta_2 = [-0.3, -0.0075, 0.2, -0.05, -0.015, -0.001, 0.02, -2]$: Coefficients for the second dataset.
- $\epsilon_1 \sim \mathcal{N}(0, 0.2^2)$: Noise for the first dataset.
- $\epsilon_2 \sim \mathcal{N}(0, 0.2^2)$: Noise for the second dataset.
- Seed for reproducibility: 42.
- Proportion of outliers introduced: 20%.
- Features corrupted varies.

**Viability Study I. Summary of the benchmarks**. Below we describe the key benchmarks.

**Benchmark 1. New model retraining**. We use the Drift Detection Method (DDM) to monitor changes in the data distribution and retrain the model when a drift is detected. The procedure includes:

1. Split the test data into multiple batches.
2. Train the model on the initial training dataset.
3. For each batch in the test set:
   - Predict the outcomes and calculate the accuracy.
   - Update the drift detector with the prediction error (1 - accuracy).
   - If drift is detected:
     - Retrain the model on the most recent batch.

**Benchmark 2. Ensemble method**. This algorithm uses an ensemble of models to improve robustness against data shifts. It combines the predictions of multiple models, each trained on different segments of the data. The procedure involves:

1. Initialize an ensemble with a single model trained on the initial training dataset.
2. Split the test data into multiple batches.
3. For each batch in the test set:
   - Aggregate predictions from all models in the ensemble, weighted by their current accuracies.
   - Make final predictions based on the weighted aggregation.
   - Calculate the accuracy and update the drift detector with the prediction error.
   - If drift is detected:
     - Train a new model on the current batch and add it to the ensemble.
     - Update the weights of all models based on their accuracies.

This method maintains a diverse set of models that can adapt to different aspects of the data distribution, enhancing overall performance and stability.

**Benchmark 3. Partial updating**. The model is retrained using a sliding window of the most recent data batches. This allows continuous adaptation to recent changes in the data distribution. The steps are:

1. Split the test data into multiple batches.
2. Train the model on the initial training dataset.
3. Maintain a buffer to store the most recent batches.
4. For each batch in the test set:
   - Predict the outcomes and calculate the accuracy.
   - Update the buffer with the current batch.
   - If the buffer exceeds a predefined size (window size), remove the oldest batch.
   - Retrain the model using the data in the buffer.

**Our method. $\mathcal{H}$-LLM**. In this example, we use $\mathcal{H}$-LLM to identify corrupted columns and values and identify whether they need removal. The overall setup is as follows:

1. Split the test data into multiple batches.
2. Train the model on the initial training dataset.
3. Maintain buffers to store the most recent batches and a backtesting window.
4. For each batch in the test set:
   - Predict the outcomes and calculate the accuracy.
   - Update the buffers with the current batch.
   - Update the drift detector with the prediction error.
   - If drift is detected:
     - Use the self-healing mechanism to inspect the most recent and previous batches.
     - Propose multiple adaptation strategies
     - Select the best adaptation strategy on a backtesting window.
     - Retrain the model on the inspected and backtesting data to recover from the detected drift.

In all cases, the optimal strategy was removing a corrupted batch of data, where the amount of corrupted values or their extent varied.

**Comments on the experimental setup of viability study I**. The goal of this setup is to showcase that blindly retraining the model or using pre-determined actions is not necessarily optimal. In this case, the strategy required is to understand that the model requires full re-training *and* some values have been corrupted which require careful dealing, such as adjustments or removal.

### C.2.2   Viability Studies III - VI

.

**Viability Study III**. We employ the Drift Detection Method (DDM) and vary the sensitivity parameter indicated on the x-axis. We then calculate the recovery time — how much time it takes to detect the shift—, as well as the post-intervention accuracy. As discussed in the main paper, this is purely determined by the DDM. For each detected drift, we fully run $\mathcal{H}$-LLM to detect issues and propose adaptation strategies that are tested on a backtesting window. If none of them beat the performance of the current model, the existing model $f$ is deployed.

**Viability Study IV**. We evaluate how well self-healing systems identify the root causes of problems. We corrupt a proportion of observations (*corruption coefficient*) by multiplying their values by a factor (*outlier factor*) and see if the $\mathcal{H}$-LLM detects issues related to these factors. We output a probability distribution over diagnoses of which variable is corrupted. Knowing the true corrupted variable, we measure the difference between the distributions using KL-Divergence, with lower values indicating better matches between true and estimated corruption. A uniform diagnosis baseline represents random guessing. Here is an example of what it means for the "true probabilities" to be corrupted when the corrupted column is "Age".

```
1  true_probabilities = {'Age': 1,
2  'HbA1c': 0,
3  'FastingGlucose': 0,
4  'BMI': 0,
```

```
5   'BloodPressure': 0,
6   'Cholesterol': 0,
7   'Insulin': 0,
8   'PhysicalActivity': 0}
```
Code Listing 11: An example of true corrupted probabilities

Recall that $\mathcal{H}$-LLM produces normalized probability guesses, as shown in Sec. B.3. Therefore, the obtained predicted guesses of which variable is corrupted in this setup looks as follows:

```
1   predicted_probabilities = {'Age': 0.125,
2   'HbA1c': 0.125,
3   'FastingGlucose': 0.125,
4   'BMI': 0.125,
5   'BloodPressure': 0.125,
6   'Cholesterol': 0.125,
7   'Insulin': 0.125,
8   'PhysicalActivity': 0.125}
```
Code Listing 12: An example of predicted corrupted probabilities

When the corruption coefficient is higher, the output looks as follows:

```
1   predicted_probabilities = {'Age': 0.4,
2   'HbA1c': 0.2,
3   'FastingGlucose': 0.15,
4   'BMI': 0.05,
5   'BloodPressure': 0.05,
6   'Cholesterol': 0.05,
7   'Insulin': 0.05,
8   'PhysicalActivity': 0.05}
```
Code Listing 13: An example of true corrupted probabilities

Therefore, the KL divergence is computed between these two probability distributions. The KL is the highest when the outputted probability distribution is uniform (first example) and the lowest when it perfectly matches the reference/true probability distribution. It has been shown that with certain techniques, LLMs can generally output calibrated confidence scores or probabilities [75].

The reason why the KL-divergence decreases is because the predicted probabilties put greater relative value on the true corrupted value (i.e. the "Age" column in this example) as (i) the outlier factor increases and as (ii) the percent of values corrupted increase.

**Viability Study V**. We study the sensitivity of SHML adaptation policies by examining how well actions perform based on (i) the number of corrupted values and (ii) the size of the backtesting dataset. Fig. 6 shows this relationship. The corruption coefficient is described in the overall experimental setup. The size of the backtesting window is the size of the dataset used to evaluat the proposed actions. Recall that $\mathcal{H}$-LLM has three adaptation actions in place: (i) generic; (ii) filtering corrupted data slices; and (iii) training slice-specific models (Appendix B). For this experiment, we focus on actions proposed by the second adaptation strategy: filtering corrupted data slices. Each adaptation action is an identified data slice by $\mathcal{H}$-LLM that might be corrupted, the removal of which might improve performance. The following is an example of proposed adaptation actions by the removal of the following queries (each query is a separate candidate adaptation action):

```
1   ['FastingGlucose > 376.145108',
2   'Insulin > 320.642677',
3   'HbA1c > 21.553946',
4   'Age > 187.805319',
5   'BMI > 93.998780',
6   'BloodPressure > 452.899287',
7   'Cholesterol > 757.675355',
8   'PhysicalActivity > 11.314583',
9   '(HbA1c > 21.553946) & (FastingGlucose > 376.145108)',
10  '(Age > 187.805319) & (BMI > 93.998780)',
11  '(BloodPressure > 452.899287) & (Cholesterol > 757.675355)',
12  '(Insulin > 320.642677) & (PhysicalActivity > 11.314583)',
```

```
13  '(HbA1c > 21.553946) & (FastingGlucose > 376.145108) & (Age >
        187.805319)',
14  '(BMI > 93.998780) & (BloodPressure > 452.899287) & (Cholesterol >
        757.675355)',
15  '(Insulin > 320.642677) & (PhysicalActivity > 11.314583) & (HbA1c >
        21.553946)',
16  '(FastingGlucose > 376.145108) & (Age > 187.805319) & (BMI >
        93.998780)',
17  '(BloodPressure > 452.899287) & (Cholesterol > 757.675355) & (Insulin
        > 320.642677)',
18  '(PhysicalActivity > 11.314583) & (HbA1c > 21.553946) & (
        FastingGlucose > 376.145108)',
19  '(Age > 187.805319) & (BMI > 93.998780) & (BloodPressure >
        452.899287)']
```

Code Listing 14: Proposed adaptation actions by removing candidate corrupted slices

Such actions are proposed for each range of values corrupted and evaluated accordingly.

**Viability study VI**. We study the importance of the testing component (Eq. 7) by evaluating $\mathcal{H}$-LLM suggested actions with and without the testing phase (backtesting window) and comparing their accuracies. Fig. 7 shows this relationship. The action with the backtesting window is the action which has received the highest empirical performance on the backtesting window. In contrast, the action proposed by "no backtesting window" is the action that is selected as the most likely one by $\mathcal{H}$-LLM without any empirical validation. "Most likely" implies that after a few iteration loops, this was the action that was listed as the first action to perform. This showcases the usefulness of having a way to filter out actions with some specific actions. We mimic the setup from study IV where each action is a specific subgroup to filter out to achieve better performance due to the corrupted nature of the data.

## C.3   Other experimental details

We note that all experiments were performed using two compute resources: a server with NVIDIA RTX A4000 GPU and 18-Core Intel Core i9-10980XE, as well as an Apple M1 Pro 32GB RAM. We exemplify $\mathcal{H}$-LLM with GPT-4 via an API.

# D Extended experiments

This section provides a few additional experiments or more detail regarding the experiments presented in the main paper.

## D.1 Monitoring

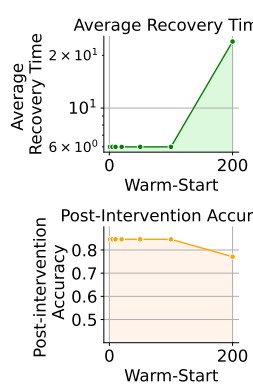

**Setup**. We vary the warm-star criterion within drift detection methods to evaluate the recovery time and post-intervention accuracy of $\mathcal{H}$-LLM. The warm start parameter is the minimum number of samples required to conclude that a drift has been detected and trigger re-training or self-healing.

**Discussion**. Fig. 8 showcases the relationship between the warm-start parameter and the average recovery tiem and post-intervention accuracy. You see the massive increase in average covery time that jumps when the warm-start is set at a relatively high threshold. This results from a drift detection algoritihm detecting a false positive drift just before the actual drift. However, given the wwarm-star parameter, there was a significant delay in re-triggering the self-healing system. This suggests self-healing systems benefit from lower warm-start parameters in case the drift detection algorithms are sensitive to false positives. This corresponds with a relative drop in the post-intervention accuracy because of the longer time it took to trigger self-healing.

Figure 8: Adaptation strategies of different methods in response to three shifts.

**Takeaway**. Self-healing systems benefit from lower warm-start parameters in case drift detection systems are sensitive to false positive drifts.-intervention accuracy with smaller thresholds.

## D.2 Diagnosis

**Setup**. In this experiment, instead of corrupting a single variable which is responsible for model degradation, we corrupt $n$ variables to evaluate how well $\mathcal{H}$-LLM can diagnose multiple corrupted values at once. With each corrupted columns, the true corrupted probability changes. For instance, if there are four columns and there is a single corrupted column, the true corruption vector is [1, 0, 0, 0]. If there are four corrupted columns, then it is [0.25, 0.25, 0.25, 0.25]. We use these probabilities and compare them to the corruption probabilities outputted by $\mathcal{H}$-LLM. This is shown in Fig. 9.

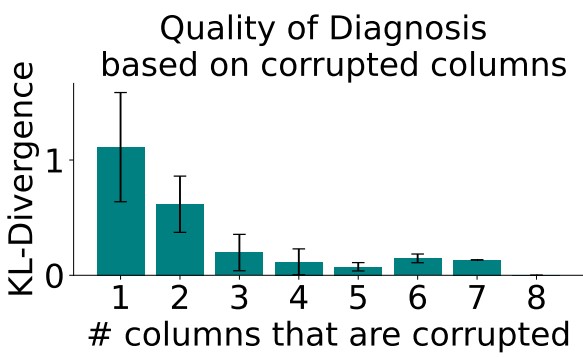

Figure 9: The qualtiy of diagnosis based on n columns. Lower is better

**Discussion**. This showcases that the more columns are corrupted, the better the predictive diagnosis becomes. For instance, once all columns are corrupted, $\mathcal{H}$-LLM outputs a uniform diagnosis because it has no information given the evidence observed. This exactly corresponds to the true corruption probability, outputting a KL of 0. We notice that the KL generally decreases with the number of corrupted columns for this reason.

**Takeaway**. Greater uncertainty results in more uniform diagnosis. However, less uncertainty can make it difficult to directly pinpoint the exact cause, causing more uncertainty.

### D.3 Adaptation experiment

This section expands on the adaptation experiments by providing more variables and values by corruption coefficient and the number of columns corrupted.

Table 7: Accuracy based on the number of corrupted columns, where 5% of values given a selected column are corrupted on a shifted dataset with a number of corrupted values. Higher is better. (with a corruption coefficient of 0.05)

|  | 1 | 2 | 3 | 4 | 5 | 6 | 7 | 8 |
|---|---|---|---|---|---|---|---|---|
| No retraining | 0.43 ± 0.02 | 0.44 ± 0.02 | 0.44 ± 0.02 | 0.44 ± 0.02 | 0.45 ± 0.02 | 0.45 ± 0.02 | 0.45 ± 0.02 | 0.45 ± 0.02 |
| Partially Updating | 0.72 ± 0.02 | 0.71 ± 0.02 | 0.70 ± 0.02 | 0.69 ± 0.02 | 0.68 ± 0.02 | 0.67 ± 0.02 | 0.65 ± 0.02 | 0.54 ± 0.06 |
| New model training | 0.71 ± 0.02 | 0.70 ± 0.02 | 0.69 ± 0.02 | 0.69 ± 0.02 | 0.68 ± 0.02 | 0.67 ± 0.02 | 0.64 ± 0.02 | 0.50 ± 0.02 |
| Ensemble Method | 0.71 ± 0.02 | 0.70 ± 0.02 | 0.69 ± 0.02 | 0.69 ± 0.02 | 0.68 ± 0.02 | 0.67 ± 0.02 | 0.64 ± 0.02 | 0.50 ± 0.02 |
| $\mathcal{H}$-LLM | 0.95 ± 0.01 | 0.93 ± 0.01 | 0.90 ± 0.02 | 0.87 ± 0.01 | 0.84 ± 0.02 | 0.79 ± 0.02 | 0.77 ± 0.02 | 0.68 ± 0.02 |

Table 8: Accuracy based on the number of percent of corrupted value within a given column (with three corrupted columns with three corrupted columns)

|  | 0.01 | 0.02 | 0.05 | 0.1 | 0.2 | 0.3 | 0.5 |
|---|---|---|---|---|---|---|---|
| No retraining | 0.43 ± 0.02 | 0.44 ± 0.02 | 0.44 ± 0.02 | 0.45 ± 0.02 | 0.46 ± 0.02 | 0.48 ± 0.02 | 0.49 ± 0.03 |
| Partially Updating | 0.74 ± 0.03 | 0.72 ± 0.02 | 0.70 ± 0.02 | 0.66 ± 0.02 | 0.62 ± 0.02 | 0.57 ± 0.02 | 0.52 ± 0.03 |
| New model training | 0.77 ± 0.02 | 0.74 ± 0.02 | 0.69 ± 0.02 | 0.66 ± 0.02 | 0.61 ± 0.02 | 0.55 ± 0.02 | 0.51 ± 0.03 |
| Ensemble Method | 0.77 ± 0.02 | 0.74 ± 0.02 | 0.69 ± 0.02 | 0.66 ± 0.02 | 0.61 ± 0.02 | 0.55 ± 0.02 | 0.51 ± 0.03 |
| $\mathcal{H}$-LLM | 0.95 ± 0.01 | 0.94 ± 0.01 | 0.90 ± 0.02 | 0.82 ± 0.02 | 0.70 ± 0.02 | 0.57 ± 0.02 | 0.52 ± 0.03 |

### D.4 Effects of Self-Healing across corruption levels

We systematically analyze how self-healing effectiveness varies with corruption levels across our five datasets (Airlines, Poker, Weather, Electricity, and Forest Type). For each dataset, we vary both the corruption value $\tau$ and the number of corrupted columns $k$, measuring accuracy with and without the self-healing mechanism. Figure 10 shows that self-healing's impact grows with corruption severity. Specifically, as either $\tau$ or $k$ increases, the gap between baseline and self-healed performance widens. This pattern holds consistently across all datasets, though with varying magnitudes. These results demonstrate that self-healing becomes more crucial as data degradation becomes more severe, providing a safety mechanism for maintaining model performance under challenging conditions.

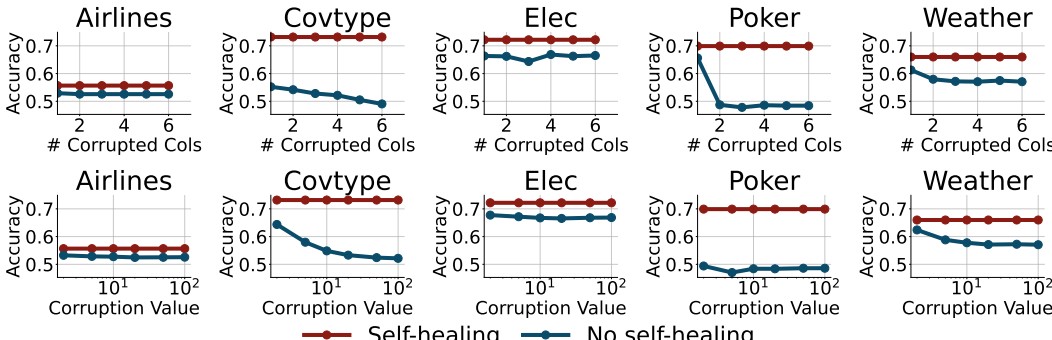

Figure 10: Effects of self-healing for five datasets as we vary the number of corrupted columns and the corruption value. Self-healing consistently identifies corrupted columns at test time. This typically becomes more important as the corruption level increases (either by corruption value or number of corrupted columns). Baseline is not implementing a self-healing mechanism upon drift detection.

### D.5 Extended benchmarks

We extend our comparison on the diabetes prediction task to include additional adaptation methods and adaptive algorithms. Table 9 presents results from ten different approaches, including standard adaptations (no retraining, partial updates, new model training, ensemble methods), streaming-specific algorithms (ADWIN Bagging, Hoeffding Tree), and our SHML approach.

| Method | Adaptations | | | | | Algorithms | | | | SHML |
|---|---|---|---|---|---|---|---|---|---|---|
| | No retraining | Partially updating | New model training | Ensemble method | Airstream | ADWIN Bagging | Hoeffding Tree | Adaptive Voting | Adaptive RF | Self-Healing ML |
| Accuracy | 0.52 ± 0.16 | 0.65 ± 0.13 | 0.65 ± 0.12 | 0.64 ± 0.12 | 0.59 ± 0.13 | 0.68 ± 0.10 | 0.70 ± 0.10 | 0.62 ± 0.11 | 0.69 ± 0.09 | **0.76 ± 0.08** |

Table 9: Accuracies of various adaptations on the original diabetes dataset setup in the paper.

The results show that while specialized streaming algorithms (e.g., Hoeffding Tree at 0.70 accuracy) outperform basic adaptations, they still fall short of SHML's performance (0.76 accuracy).

## D.6 Component-wise Ablation Analysis

To understand the importance of each SHML component, we conduct an ablation study by systematically removing each component and observing the impact. Table 10 shows the results of this analysis.

| Ablation | Accuracy (%) | Takeaway |
|---|---|---|
| Baseline (no self-healing) | 52 | Accuracy is worse without self-healing |
| Full (full self-healing) | 76 | Self-healing improves accuracy over baseline. |
| No monitoring | 52 | Monitoring is required to trigger the SHML system. $\mathcal{H}$-LLM was not triggered and no actions were proposed. |
| No diagnosis | 52 | Diagnosis is required for proposing sensible actions. Defaults to non-sensical actions. |
| No actions | 52 | Actions could not be implemented because they were not proposed, defaults to no behavior. |
| No testing | 62 | Actions chosen but not tested against empirical data. A suboptimal action was chosen. |

Table 10: Ablation study results for $\mathcal{H}$-LLM. We systematically remove one component of the system and inspect its outputs. The takeaway represents our qualitative evaluation.

The ablation reveals that each component is crucial for effective self-healing. Removing monitoring (52% accuracy) prevents the system from triggering adaptation. Without diagnosis, the system proposes non-sensical actions, leading to baseline performance. Removing action generation or testing similarly degrades performance to baseline levels, though testing removal shows slightly better performance (62%) as some reasonable actions are still attempted, albeit without proper validation.

This analysis empirically validates our framework's design, showing that effective self-healing requires all four components working in concert.

## D.7 Model agnosticism

We evaluate SHML's effectiveness across ten different ML models to demonstrate its model-agnostic nature. Table 11 shows results for models ranging from simple (e.g., Decision Trees) to complex (e.g., XGBoost), comparing various adaptation strategies. SHML consistently outperforms baseline approaches across all model types, with improvements ranging from 11 percentage points (Naive Bayes) to 31 percentage points (LDA). This consistent improvement demonstrates that SHML's benefits are not tied to any particular model architecture but rather stem from its ability to reason about and address degradation causes.

| Method | DecisionTree | KNN | LDA | LogisticRegression | MLP | NaiveBayes | Perceptron | RandomForest | SGD | XGBoost |
|---|---|---|---|---|---|---|---|---|---|---|
| Baseline (No retraining) | 0.63 ± 0.05 | 0.51 ± 0.03 | 0.47 ± 0.03 | 0.49 ± 0.02 | 0.63 ± 0.04 | 0.51 ± 0.03 | 0.49 ± 0.01 | 0.63 ± 0.05 | 0.47 ± 0.03 | 0.67 ± 0.05 |
| Sliding Window | 0.63 ± 0.05 | 0.51 ± 0.03 | 0.47 ± 0.03 | 0.49 ± 0.02 | 0.66 ± 0.03 | 0.51 ± 0.03 | 0.49 ± 0.01 | 0.70 ± 0.05 | 0.47 ± 0.03 | 0.67 ± 0.05 |
| Drift Detection (DDM) | 0.63 ± 0.05 | 0.51 ± 0.03 | 0.47 ± 0.03 | 0.49 ± 0.02 | 0.64 ± 0.04 | 0.51 ± 0.03 | 0.49 ± 0.01 | 0.66 ± 0.05 | 0.47 ± 0.03 | 0.67 ± 0.05 |
| Ensemble with DDM | 0.63 ± 0.05 | 0.51 ± 0.03 | 0.47 ± 0.03 | 0.49 ± 0.02 | 0.65 ± 0.05 | 0.51 ± 0.03 | 0.49 ± 0.01 | 0.65 ± 0.07 | 0.47 ± 0.03 | 0.67 ± 0.05 |
| $\mathcal{H}$-LLM | **0.70 ± 0.04** | **0.73 ± 0.05** | **0.77 ± 0.04** | **0.76 ± 0.04** | **0.78 ± 0.05** | **0.62 ± 0.02** | **0.68 ± 0.09** | **0.72 ± 0.04** | **0.75 ± 0.04** | **0.71 ± 0.04** |

Table 11: Comparison of various methods across different ML models on the weather dataset (setup above), where features are corrupted at test time. Results show mean accuracy ± standard deviation.

# E  Optimal diagnosis

Here, we prove that under the stated assumptions, the optimal diagnosis has zero entropy.

**Proposition 3.** *Under Assumption 1, the optimal diagnosis $\zeta^*$ has a zero entropy, i.e., $\mathbb{H}(\zeta^*) = 0$.*

*Proof.* By Definition 2,

$$\zeta^* = \arg\min_{\zeta \in \Delta(\mathcal{Z})} \mathbb{E}_{a \sim \pi(\cdot|\zeta)}[R(a)] \tag{12}$$

As $\mathcal{A}$ is finite, we write the expected value as follows.

$$\mathbb{E}_{a \sim \pi(\cdot|\zeta)}[R(a)] = \sum_{a \in \mathcal{A}} R(a)\pi(a|\zeta) \tag{13}$$

By Assumption 1, this can be rewritten as:

$$\sum_{a \in \mathcal{A}} R(a)\left(\sum_{z \in \mathcal{Z}} \pi(a|z^\dagger)\zeta(z)\right) \tag{14}$$

We change the order of summation to arrive at the following.

$$\sum_{z \in \mathcal{Z}} \zeta(z) \sum_{a \in \mathcal{A}} R(a)\pi(a|z^\dagger) \tag{15}$$

The inner sum can now be rewritten as an expectation.

$$\sum_{z \in \mathcal{Z}} \zeta(z)\mathbb{E}_{a \sim \pi(\cdot|z^\dagger)}[R(a)] \tag{16}$$

Thus we can rewrite the minimization problem as follows.

$$\zeta^* = \arg\min_{\zeta \in \Delta(\mathcal{Z})} \sum_{z \in \mathcal{Z}} \zeta(z)\mathbb{E}_{a \sim \pi(\cdot|z^\dagger)}[R(a)] \tag{17}$$

Let $z^* \in \mathcal{Z}$ such that

$$z^* \in \arg\min_{z \in \mathcal{Z}} \mathbb{E}_{a \sim \pi(\cdot|z^\dagger)}[R(a)] \tag{18}$$

Then

$$\begin{aligned}
\sum_{z \in \mathcal{Z}} \zeta(z)\mathbb{E}_{a \sim \pi(\cdot|z^\dagger)}[R(a)] &\geq \sum_{z \in \mathcal{Z}} \zeta(z)\mathbb{E}_{a \sim \pi(\cdot|(z^*)^\dagger)}[R(a)] \\
&= \mathbb{E}_{a \sim \pi(\cdot|(z^*)^\dagger)}[R(a)] \\
&= \sum_{z \in \mathcal{Z}} (z^*)^\dagger(z)\mathbb{E}_{a \sim \pi(\cdot|z^\dagger)}[R(a)]
\end{aligned} \tag{19}$$

Therefore

$$\zeta^* = (z^*)^\dagger \tag{20}$$

and by the definition of entropy and $(z^*)^\dagger$ we get

$$\mathbb{H}(\zeta^*) = 0 \tag{21}$$

$\square$

