# OpenReview forum: "Self-Healing Machine Learning: A Framework for Autonomous Adaptation in Real-World Environments"
_NeurIPS.cc/2024/Conference — NeurIPS 2024 poster_

### Official Review · Reviewer_xkmy · 2024-07-11

**Soundness:** 3
**Presentation:** 3
**Contribution:** 3
**Rating:** 6
**Confidence:** 5

**Summary:**

The paper titled "Self-Healing Machine Learning: A Framework for Autonomous Adaptation in Real-World Environments" introduces the concept of self-healing machine learning (SHML). This framework aims to address performance degradation in machine learning models due to distributional shifts. Unlike traditional concept drift adaptation methods, SHML focuses on diagnosing the reasons for model degradation and proposing targeted corrective actions. The paper presents a theoretical foundation for SHML, an algorithm (H-LLM) that uses large language models for diagnosis and adaptation, and empirical evaluations demonstrating the effectiveness of the approach.

**Strengths:**

1. The SHML framework is a novel approach to handling model degradation by diagnosing and addressing the root causes, rather than using reason-agnostic methods.
2. The paper provides a solid theoretical foundation for SHML and demonstrates its practical viability through empirical evaluations.
3. The concepts are well-explained, and the structure of the paper is logical and easy to follow. Figures and tables enhance the understanding of the proposed methods.
4. SHML has significant potential in high-stakes applications where maintaining optimal model performance is critical, such as healthcare and finance.

**Weaknesses:**

1. While the paper demonstrates promising results, the empirical evaluation is limited to a simulated diabetes prediction task. Additional experiments in diverse real-world environments would strengthen the claims.
2. The comparison with existing concept drift adaptation methods is not exhaustive. A broader set of baseline comparisons would provide a clearer picture of the advantages and limitations of SHML.
3. The paper lacks detailed information on the implementation of H-LLM, especially regarding the practical challenges of deploying such a system in real-world scenarios.
4. Although the authors mention the availability of the code upon acceptance, more details on the experimental setup and data used would help in reproducing the results.

**Questions:**

1. Can you provide more detailed results and analysis of SHML in a wider range of real-world environments beyond the simulated diabetes prediction task?
2. How does SHML compare to other state-of-the-art concept drift adaptation methods in terms of computational overhead and sample efficiency?
3. Can you elaborate on the specific contributions of each component of the SHML framework to the overall performance improvement?

**Limitations:**

The authors acknowledge the limitations of SHML, including the challenges in accurately diagnosing the root causes of performance degradation and the potential computational overhead of the approach. They also discuss the need for sufficient parallel processing capacity to handle the increased demands of multiple diagnostic and adaptation actions. The broader societal impact of SHML, including potential misuse in high-stakes applications, is also briefly addressed.

---

> ### Author Rebuttal · Authors · 2024-08-07
>
> Dear reviewer xkmy,
>
> Thank you for taking the time to review our paper. We're happy you think that SHML is a novel approach with a solid theoretical foundation and that has significant potential in high-stakes applications.
>
> To address your concerns, we've expanded our evaluation with **five new datasets**, **additional SHML effectiveness evaluation**, added a study with **six new benchmarks**, evaluated SHML across **10 different ML models**, and performed  **a qualitative ablation study**. We hope this information provides reasons to consider increasing your score. Our responses (A-H) follow:
>
> ---
> # (A) More datasets
>
> While our paper's main contribution is the SHML theoretical framework, with viability studies as proof-of-concept, we agree more real-world examples improve the paper. **We've added five new datasets** common in concept drift adaptation, varying corruption levels at test time to evaluate adaptation methods. Please find this in **Table 1 in the response pdf**.
>
> **Takeaway**: $\mathcal{H}$-LLM improves upon reason-agnostic baselines in five real-world datasets when the reason for performance drop can be diagnosed, even in test-time scenarios. This highlights the potential of self-healing ML.
>
> We'll include these results with more experimental details in the updated paper.
>
> ---
> # (B) Greater empirical analysis of self-healing ML
>
> To address your concern about limited empirical analysis, **we've conducted an additional viability** study across the five datasets. We vary problem difficulty (corruption level) at test time and evaluate self-healing benefits (**Fig. 1 in the pdf**).
>
>  **Takeaway**: $\mathcal{H}$-LLM improves performance relative to baselines across five real-world datasets, with greater self-healing effects at higher corruption levels.
>
> We'll include these results with more experimental details in the updated paper.
>
> ---
> # \(C) Limited benchmarks
>
> We emphasize that our viability studies aim to show that any reason-agnostic strategy will perform poorly whenever it is important to understand the reason for performance degradation. This is because such strategies do not directly address the root cause.
>
> That said, we've extended our evaluation benchmarks. **We've run an additional viability study with five benchmarks and four adaptive algorithms** common in concept drift adaptation (**Table 2 in the response pdf**).
>
> **Takeaway**: $\mathcal{H}$-LLM consistently outperforms adaptation methods and adaptive algorithms that fail to address test-time corruption, demonstrating self-healing ML's feasibility in high-stakes environments where understanding model degradation reasons is important.
>
> Additionally, we run one more viability study to evaluate whether these gains are consistent across different ML models. We show that $\mathcal{H}$-LLM consistently outperforms other adaptation methods across 10 different ML models (shown in **Table 4 in the response pdf**), highlighting the model-agnostic nature of the framework.
>
> ---
> # (D) Implementation of $\mathcal{H}$-LLM
>
> We're glad you're interested in $\mathcal{H}$-LLM's practical implementation. Besides the main text, implementation details are in the appendix: components (**Appendix B.1.**), prompt templates (**Appendix B.2**), outputs (**Appendix B.3**) and viability studies (**Appendix C.2.**). That said, we acknowledge it might be difficult to find this information.
>
> **Actions taken**: Move input/output examples and prompt structures from appendix to main text.
>
> ---
> # (E) Practical challenges of deploying self-healing systems
>
> We're excited that you're interested in the practical deployment of self-healing systems. We answer this exact concern in **Section 5.1.** titled "unique challenges of building self-healing systems". We agree that the naming of this section could be improved.
>
> **Actions taken**: We will rename section 5.1. to *Practical challenges of deploying self-healing systems* to improve clarity. We will also expand the discussion in the appendix.
>
> ---
> # (F) More details on experimental setup and data used
>
> In the camera-ready version, we'll use the additional page to move some of the experimental details and dataset information described in **Appendix C** to the main paper.
>
> ---
> # (G) SHML comparison to other concept drift adaptation methods
>
> **Computational overhead**. SHML methods have larger overhead than reason-agnostic approaches due to the self-healing system (LLM pipeline) identifying model failure reasons. Practically, it takes 20-40 seconds to implement a full pipeline and correct a model upon drift detection. This overhead is negligible for real-world systems given the benefits. Overhead may vary across systems.
>
> **Sample efficiency**. No differences exist as failure detection doesn't depend on sample size, but on self-healing pipeline complexity.
>
> **Actions taken**: Add an appendix discussing computational details for SHML systems.
>
> ---
> # (H) Contribution of each component
>
> **We've performed an additional qualitative ablation study**, running a self-healing ML system and iteratively removing one component, inspecting its output (**Table 3 in supplementary pdf**).
>
> **Takeaway**: This study illustrates that all components are required for the self-healing ML system to work. Having said this, we see the robustness of self-healing ML as a big research agenda for future work and hope this spurs research in this area.
>
> ---
> # Thank you
>
> We believe these changes should greatly enhance the paper's contribution and improve its clarity. In addition to the clarifications which will be included in the camera-ready version, we have added **five new datasets**, **additional SHML effectiveness evaluation**, evaluation across **10 ML models**, a study with **six new benchmarks**, and **a qualitative ablation study**.
>
> Thank you for your help.
>
> If we addressed your concerns, we hope you consider significantly revising your assessment of the paper's impact and the corresponding evaluation for NeurIPS. ☺️

---

> ### Author Response · Authors · 2024-08-11
>
> Dear reviewer xkmy,
>
> We thank you once again for the effort put into reviewing our paper. As there are only a couple working days left in the discussion period, we would like to ask if our response has satisfied your concerns. If so, we hope you consider revising your assessment of the paper's impact and the corresponding evaluation for NeurIPS. If any concerns remain, we are happy to discuss them further here.

---

> > ### Comment · Reviewer_xkmy · 2024-08-11
> >
> > Hello,
> >
> > Sorry for my late response. I am satisfied with your answers, therefore I am increasing my assessment..
> >
> > Regards,
> > Shadab

---

> > > ### Author Response · Authors · 2024-08-12
> > >
> > > Dear reviewer xkmy,
> > >
> > > We are glad our response was helpful and would like to thank you for raising your score! Thanks again for your time and suggestions which have helped us to improve the paper.
> > >
> > > Regards
> > >
> > > Paper Authors

---

### Official Review · Reviewer_vweT · 2024-07-12

**Soundness:** 3
**Presentation:** 3
**Contribution:** 3
**Rating:** 7
**Confidence:** 4

**Summary:**

This paper proposes a new concept of self-healing machine learning, or SHML. The idea is based on understanding and addressing the reasons of performance drops in ML systems, thereby going beyond most common approaches that are labelled as reason-agnostic. The approach is based on a pipeline well illustrated in Fig 2 that includes four steps: monitoring, diagnosis, adaptation and testing. Formal definitions are provided. Experimental support is provided to demonstrate some aspects of the new idea and its potential advantages.

**Strengths:**

I have identified the following strength:
- The paper starts from an interesting assumption that the reasons for performance drop should be considered and investigated.
- The paper is well organised and easy to read.
- The method is described with precise formalism.

**Weaknesses:**

I have identified the following weaknesses:
1. While the main assumption is move on from reason-agnostic methods to a system that attempts to understand the reasons, the paper does not offer a good classification of such reasons, mainly because it main a meta-level approach, in which reasons are to be learned. This could be less effective than pre-defining monitoring and diagnosis methods according to known causes.
2. As a follow-up from the previous point, the diagnostic part that generates hypotheses via a LLM may or may not be correct and the adaptation may or may not be effective. I find it hard to get a feeling of how well the system would work on a large set of benchmarks and problems from the evidence and experiments presented in the paper.
3. Most of the interesting details of the paper are actually in the appendix.
4. I disagree that this is the first non-reason-agnostic method, as the authors overview and list other approaches that do focus on the causes of performance drop. I agree that this may be the first approach to tackle the issue comprehensively. Nevertheless, the novelty, knowledge gap, and improvements of the approach with respect to existing approaches is not well summarised in the introduction/related work, which is rather short and it refers to the appendix for further detail.

**Questions:**

1. Is it possible to define a medium to large set of conditions in known benchmarks that could be use to extensively test the SHML?
2. One challenge in a system with multiple sequential steps is that the failure of one component affects all the following. E.g. failure in diagnosis will affect adaptation and testing.  How can we assess how well the system work when failure in one component compromises the correct functioning of others?

**Limitations:**

I find that the limitation part (before the conclusion and in 5.2)  is too short . Yes, I agree that the correct identification of the root causes of performance drop is challenging in real work scenarios, but the issue needs to be expanded and discussed more. For example, why is it difficult? Are some reasons that are more difficult than others to identify? Monitoring could also be challenging. Adaptation could be challenging.

The impact is sufficient covered.

---

> ### Author Rebuttal · Authors · 2024-08-07
>
> Dear R-vweT,
>
> we're glad you think our paper is important and easy to read.
>
> ---
> # (A) Classification of reasons for performance drops
>
> You're right that we don't provide an exhaustive classification of reasons for performance drops. We aim SHML to be a meta-level approach which flexibly adapts to various scenarios.
>
> A meta-level approach offers two main advantages: 1) It allows the system to adapt to unforeseen scenarios and new types of model degradation not captured by pre-defined lists. 2) It flexibly generates multiple hypotheses to explain the same model degradation, as different explanations may account for the same issue (illustrated in **Table 2** of the main paper).  Also note that SHML could incorporate pre-defined causes into the framework, as these represent deterministic if-else diagnosis functions and would result in a less complex policy function $\pi$ than explored in the paper.
>
> **Actions taken**: Include the classification of performance drops in the discussion.
>
> ---
>
> # (B) How well the system works on large set of benchmarks
>
> We introduce self-healing ML as a framework for understanding model degradation. Our goal is to demonstrate its viability rather than provide extensive benchmark comparisons.
>
> That said, to address your concern directly, we've conducted additional viability studies:
>
> - We compare $\mathcal{H}$-LLM, with baseline approaches on **five real-world datasets** by introducing corruption at test time (**Table 1 in supplementary pdf**). **Takeaway**: $\mathcal{H}$-LLM improves upon reason-agnostic baselines when the reason for performance drop can be diagnosed.
> - We show how the importance of self-healing ML varies with the problem difficulty (**Fig. 1** in the response pdf). **Takeaway**: The effects of self-healing are greater when the data corruption levels are higher.
>
> **Actions taken**: We will include a summary of this discussion in the main paper.
>
> ---
> # \(C) The effectiveness of each component
>
> We emphasize that we see robust studies of each component as non-trivial research directions, as our paper primarily aims to introduce SHML. That said, we address your concern by performing a qualitative ablation study. **Setup**: We run $\mathcal{H}$-LLM and iteratively remove one component from the system, inspecting its output. This is presented in **Table 3 in the response pdf**.
>
> **Takeaway**: We illustrate that *all components are required for the SHML system to work*.
>
> ---
> # (D) Moving items from the appendix to the main text
>
> In the camera-ready version, we'll use the additional page to expand the viability section in the main text. We'll include more details on the prompts and input/output examples of $\mathcal{H}$-LLM.
>
> ---
> # (E) Is self-healing the first reason-agnostic method?
>
> You're correct that there have been previous papers attempting to identify reasons for model degradation, discussed in **Section 2** (L83 - 90 and L105-120). To clarify our contribution, we've created a comparison table highlighting the key differences between SHML and existing approaches:
>
> | Approach                                    | Diagnosis    | Adaptation Action |
> |-|-|-|
> | Concept drift adaptation [1,2]              | n/a        | Fixed             |
> | Specialized drift handling [3,4]            | n/a        | Fixed             |
> | Distribution change attribution [5]         | Fixed     | n/a             |
> | Model failure attribution [6]               | Fixed     | n/a             |
> | Dynamic classifier selection [7]            | n/a        | Fixed          |
> | SHML (Our approach)                         | Variable     | Variable          |
>
> [1] Gama et al. (2004), [2] Lu et al. (2018), [3] Goncalves et al. (2013), [4] Alippi et al. (2013),
> [5] Budhathoki et al. (2021), [6] Zhang et al. (2022), [7] Cruz et al. (2018)
>
> **Actions taken:** We will highlight that SHML is the first framework where the diagnosis and adaptation are not fixed in advance and where the diagnosis informs the adaptation.
>
> ---
> # (F) Defining conditions in known benchmarks that could be used to extensively test SHML
>
> We agree that having an extensive set of known conditions could be extremely useful for evaluating SHML systems that would likely gain a lot of traction in the ML community. We think there are multiple conditions that could be used to simulate real-world conditions in existing benchmarks (such as data corruption, systematically changing the DGP, introducing external shocks such as covid-19) which would require significant testing and validation. We see this as promising future work.
>
> ---
> # (G) Failures in sequential systems and how that affects self-healing ML
>
> You're right that failure in one component can affect the entire system's performance.
>
> **a) Component sensitivity analysis**. To provide initial insights, we refer to the qualitative ablation study presented earlier in our rebuttal. The study showcased that each component working is required for the system's overall performance.
>
> **b) Built-in safeguards**. SHML includes mechanisms to mitigate cascading failures with the testing component (step 4). Actions that do not improve performance over baseline are discarded.
>
> We see designing fail-safe self-healing systems as a promising research direction.
>
> ---
> # (H) Expanding limitations
>
> In the camera-ready version, we'll expand the discussion of the limitation sections.
>
> 1. We'll add a paragraph discussing specific challenges with root cause identification (expanding on and moving items from section 5.1 lines 208 - 2012)
>
> 2. We'll add a paragraph discussing the challenges with choosing an appropriate action (expanding on and moving items from section 5.1 lines 213-216)
>
> 3. We'll add a challenge explaining when understanding the root cause is easy/difficult based on our experience.
>
> ---
> # Thank you
>
> You have helped us improve our paper. Given these changes, we hope you consider revising your assessment of the paper's impact and the corresponding evaluation for NeurIPS. ☺️

---

> > ### Comment · Reviewer_vweT · 2024-08-08
> >
> > I appreciate the careful response and effort to address my concern. I believe the paper has improved as a consequence, and I'm happy to increase my assessment.

---

> > > ### Author Response · Authors · 2024-08-08
> > >
> > > Dear reviewer vweT,
> > >
> > > We are glad our response was helpful and would like to thank you for raising your score! Thanks again for your time and suggestions which have helped us to improve the paper.
> > >
> > > Regards
> > >
> > > Paper Authors

---

### Official Review · Reviewer_4TYN · 2024-07-12

**Soundness:** 3
**Presentation:** 2
**Contribution:** 3
**Rating:** 6
**Confidence:** 3

**Summary:**

The paper presents a self-healing framework for machine learning models called Self-Healing Machine Learning (SHML). Unlike previous methods, SHML autonomously diagnoses the causes of model degradation and suggests corrective actions based on these diagnoses. The authors formalize SHML as an optimization problem, aiming to minimize expected risk by adapting to changes in the data-generating process (DGP).
A theoretical framework for self-healing systems is introduced, exemplified by H-LLM, which leverages large language models for self-diagnosis and self-adaptation. Empirical analyses of SHML's components demonstrate its potential and effectiveness.
The paper underscores the importance of ensuring optimal performance in algorithms used in high-stakes applications. By enabling systems to autonomously adapt to new environments, SHML aims to advance self-healing systems, benefiting both the machine learning community and society. The theoretical framework lays the groundwork for developing optimal adaptation and diagnosis methods. The authors hope this work will stimulate further theoretical developments and encourage the adoption of self-healing systems in critical fields such as medicine and finance.

**Strengths:**

From the perspective of originality, this paper lowers the barrier for others to implement adaptation actions in machine learning models. Its importance lies in addressing the growing challenge of maintaining machine learning models, especially given their increasing usage. A framework capable of diagnosing drift or degradation and autonomously solving these issues is crucial. This paper effectively addresses this need, outperforming existing approaches in both presentation and results.
The organization of the paper is good, with a well-structured presentation. The detailed inclusion of examples, definitions, assumptions, theoretical explanations, and results in the main text and appendix is thorough and effective (but the appendix is probably required to really understand)

**Weaknesses:**

The main weakness of this paper is its tendency to use overly long sentences and complex phrases, which can hinder readability and clarity. Another significant weakness lies in the viability section; many crucial details have been relegated to the appendix. This makes it difficult to fully understand and trust the framework based on the main text alone. Additionally, while the concept of "self-healing machine learning" is compelling, the section dedicated to it could be more concise and focused. Overall, improving the clarity of these sections would greatly enhance the paper's impact.

**Questions:**

1. Is the diagnostic step continuously operational during the entire period of model usage?
2. How does the performance of the framework compare to benchmarks when applied to large-scale models?

**Limitations:**

Authors mention a limitation in section "SHML’s success relies on accurate root cause identification and finding effective adaptation policies which could pose challenges in some complex, real-world settings" (Sec. 5.1).
This could be further explored.

---

> ### Author Rebuttal · Authors · 2024-08-07
>
> Dear reviewer 4TYN,
>
> Thank you for carefully reading our paper. We're glad that you think we address the problem of maintaining machine learning models autonomously and that this lowers the barriers for others. We respond to each point below.
>
> ---
> # (A) Moving information from the Appendix to the main text
>
> In the camera-ready version, we'll use the additional page to expand the viability section in the main text. We'll include more details on the prompts and input/output examples of $\mathcal{H}$-LLM. We will also highlight which design choices are most important to consider such that adopters can more easily replicate and build their own self-healing systems.
>
> ---
> # (B) Improving the clarity of Section 3 on self-healing ML
>
> We agree with you that the main section could be more concise and focused. To improve clarity, we will make the following changes:
>
> a) We'll start with a concise overview of SHML:
>
> > Self-healing machine learning is a framework for autonomously detecting, diagnosing, and correcting performance degradation in deployed ML models. It aims to maintain model performance in changing environments without constant human intervention.
>
> b) In the box "Self-Healing Machine Learning in a nutshell.", we will add a more intuitive explanation at the beginning.
>
> > SHML contains four components: (a) **monitoring**: continuous assessment of model performance; (b) **diagnosis**: identification of root causes for performance degradation; \(c) **adaptation**: suggesting possible corrective actions to take in response to degradation, and (d) **testing**: empirically evaluating the effect of actions on the model's performance. After these steps, the best action is implemented on the ML model. This is illustrated in Fig. 2.
>
> c) We will contrast SHML with traditional approaches at the end of Sec. 3.3 (lines 152-154):
>
> > The primary insight of SHML is that the best action to take in response to model degradation depends on the reason for that degradation. This contrasts with standard approaches, which often assume the best approach is independent of the degradation reason. For example, a standard drift adaptation method might continuously retrain the model, which could be suboptimal if the new dataset is corrupt.
>
> We will also make other smaller changes, such as add a clear definition of $f$ at the beginning, clarify the relationship between $f$ and the policy $\pi$, and include an illustrative example.
>
> ---
> # \(C) Question: Is the diagnostic step continuously operational?
>
> The diagnostic step is not continuously operational. It's triggered only when the monitoring component detects performance degradation, as shown in Figures 2 and 3. This design choice is due to the computational intensity of the diagnostic step. In our implementation of $\mathcal{H}$-LLM, we perform multiple language model calls to hypothesize possible model failures, propose actions and implement them. In practice, we find this loop takes about 20 seconds to a minute to finish.
>
> Continuous diagnostics might be feasible in two scenarios:
>
> a) Batch prediction settings: If your model makes predictions in large batches (e.g., every 15 minutes) rather than continuously (e.g., every second), a diagnosis step could potentially run for each batch, even without detected degradation.
>
> b) Future research could explore novel architectures where LLM-based monitoring systems run concurrently and continuously with monitoring. However, to the best of our knowledge, no such systems currently exist.
>
> **Actions taken**: We will clarify when the diagnosis is triggered in the main paper more clearly and expand on when it might be continuously operational.
>
> ---
> # (D) Question: performance with large-scale models
>
> The SHML framework is agnostic to the model $f$ (L139-141 in the main text). Therefore, self-healing ML shows superiority over benchmarks whenever understanding the reason for degradation is important, *regardless of the learner*. We can intuitively explain this with an example: suppose that a new batch of data is corrupt. Using a reason-agnostic approach, such as retraining your model on new data, is suboptimal regardless of the model used. This is because the optimal action requires to remove the corrupted data and then retrain the model on the de-corrupted data.
>
> That said, **we've conducted an additional viability study** comparing SHML's performance against other adaptation methods **for 10 popular ML models** commonly used in practice. We include models used for large-scale industry applications, e.g. XGBoost or Random Forest. We simulate real-world degradation by corrupting data at test time and evaluating the performance of each adaptation approach. The results are included in **Table 4** in the response pdf. **Takeaway**: We show that $\mathcal{H}$-LLM outperforms other adaptation methods. This showcases that self-healing ML can benefit any downstream ML model whenever understanding the reason for degradation is important (such as test time corruption).
>
>
> ---
> # (E) Limitations
>
> In the camera-ready version, we will expand our limitations.
>
> - We will move some challenges of building self-healing systems from **Section 5.1** (L206-216) to the discussion to highlight limitations.
> - We will expand on the challenges in diagnosis (L208-212) by giving an illustrative example.
> - We will expand on the challenges in adaptation (L213-216) by explaining when root cause identification might be easy/difficult.
> - We will outline research directions that can help overcome said challenges.
>
> ---
> # Thank you
>
> You have helped us improve our paper. Given these changes, we hope you consider revising your assessment of the paper's impact and the corresponding evaluation for NeurIPS. ☺️

---

> > ### Author Response · Authors · 2024-08-11
> >
> > Dear reviewer 4TYN,
> >
> > We thank you once again for the effort put into reviewing our paper. As there are only a couple working days left in the discussion period, we would like to ask if our response has satisfied your concerns. If so, we hope you consider revising your assessment of the paper's impact and the corresponding evaluation for NeurIPS. If any concerns remain, we are happy to discuss them further here.

---

> > > ### Author Response · Authors · 2024-08-13
> > >
> > > Dear reviewer 4TYN,
> > >
> > > Thank you for your thorough review of our work. As the discussion period is coming to a close, we'd like to follow up on your prior concerns.
> > >
> > > Have our responses and clarifications addressed your main questions and concerns? If so, we kindly ask that you consider increasing your score to better reflect this. If not, please let us know what specific issues remain, and we'll promptly provide additional information to resolve them.
> > >
> > > We appreciate your reconsideration and look forward to your feedback.

---

### Official Review · Reviewer_LnCp · 2024-07-13

**Soundness:** 3
**Presentation:** 3
**Contribution:** 3
**Rating:** 5
**Confidence:** 4

**Summary:**

Model performance degradation on unseen data is a classic problem. Existing approaches solve the problem through a deterministic strategy: change model, retraining, etc. This paper proposes an adaptive way to decide the action after model degradation automatically and introduces a self-healing framework. The evaluation thoroughly analyzes the intuition of SHML and its limitations.

**Strengths:**

1. Autonomous healing for model performance degradation is an important problem.
2. The automatic adaptation idea is novel and interesting.
3. The evaluation is strong and thorough, which covers the details to help readers further understand the scope and the logic of the proposed method.

**Weaknesses:**

1. The writing is hard to follow in section 3. What is the practical meaning of f? How is it related to the policy?
2. The assumption of the requirement for optimal adaptation actions is strong. Is there any practical case to support this assumption?

**Questions:**

Please refer to the weaknesses

**Limitations:**

Limitations are well-discussed in the main text.

---

> ### Author Rebuttal · Authors · 2024-08-07
>
> Dear Reviewer LnCp,
>
> Thank you for your thoughtful feedback on our work on self-healing machine learning. We appreciate your recognition of the importance and novelty of our approach. We'll address your concerns in two parts, corresponding to both weaknesses.
>
> ---
>
> # (A) Clarity of Section 3
>
> The goal of Section 3 is to describe how a deployed ML model ($f$), such as a logistic regression classifier, can be "healed" with a self-healing system ($\mathcal{H}$). In this case, *healing* refers to restoring the performance of $f$ after it drops.
>
> A natural question is: how do we know what actions we should take to heal $f$ once it drops in performance? We say that $f$ should be healed by the healing-system $\mathcal{H}$ which follows a policy $\pi$. The policy $\pi$ outputs actions $a$ (such as $a_1$: retrain a model or $a_2$: remove corrupted features) which are then implemented *onto* $f$. Therefore, $\mathcal{H}$ follows a policy $\pi$ which helps to determine optimal actions $a$ that change/modulate the deployed ML model $f$.
>
>
> We will improve the paper's writing in the following ways:
>
> a) We will add a clear definition of $f$ at the beginning:
>
> > $f$ represents the deployed machine learning model that we aim to heal. It is the function that makes the predictions on input data and whose performance we're trying to maintain and improve.
>
> In our viability studies, a logistic regression model represents $f$.
>
> b) We'll clarify the relationship between $f$ and $\pi$.
>
> > While $f$ is the model making predictions, $\pi$ is the adaptation policy --- a function that determines what actions to take to modify $f$ based on the diagnosed reasons for its performance degradation.
>
> c) We will include an illustrative example.
>
> > For instance, if $f$ is a diabetes prediction model and $\pi$ diagnoses that $f$'s performance has degraded due to concept drift, $\pi$ might suggest an action to retrain f with more recent data or to adjust feature weights.
>
> d) We will link the theory of the policy $\pi$ to the experiments.
>
> > In our viability studies with $\mathcal{H}$-LLM, the policy $\pi$ is instantiated with an LLM (GPT-4) which uses the diagnosed reasons for model failures (also achieved with an LLM) to propose concrete actions.
>
> e) We will explain how the policy impacts SHML:
>
> > Self-healing ML is formalized as "an optimization problem over a space of adaptation actions." This means we aim to find the optimal actions to take each time the model $f$ degrades. These actions are chosen by the policy $\pi$ of the self-healing system $\mathcal{H}$ (Fig. 3). For instance, two different policies $\pi_1$ and $\pi_2$ might propose different actions to take in response to $f$'s performance drop.
>
> ---
>
> # (B) Requirement for optimal adaptation actions
>
> You're correct that the assumption of optimal adaptation actions would be very strong. Optimal adaptation actions would assume that we always pick the most *optimal* action in any situation. However, this is unrealistic --- we rarely *really* know what is the optimal action to take to improve a model's performance because there are many possible reasons which could have led to the decrease in performance. So, achieving model optimality is often impossible.
>
> However, we believe there may have been a misunderstanding. Our framework doesn't require optimal actions. Rather, we aim to make more informed decisions to try to *approximate* optimal actions (which are never known in practice). We illustrate this with the model degradation example in **Sec. 3.2**.
>
> To improve the clarity, we will make the following changes:
>
> a) We will add a paragraph (lines 154-156) explaining that SHML approximates an ideal rather than achieving or assuming perfect optimality.
>
> > In real-world ML problems, it is often impossible to determine the optimal action due to the complexity of the problem. SHML attempts to approximate an ideal optimal adaptation strategy instead of achieving perfect optimality. This is done by selecting actions to take by taking into account diagnosis information instead of relying on deterministic actions (such as model retraining).
>
> b) In Section 5.2, we'll add this practical example.
>
> > Consider a scenario where both data corruption and concept drift occur simultaneously. A traditional method might simply retrain the model on new data (potentially incorporating corrupted values). In contrast, SHML would diagnose both issues and suggest a two-step adaptation strategy: (a) clean the corrupted data and (b) retrain the model on the drift-adjusted datasets. In our experiments, this has improved the model by about 18% in model accuracy, despite not necessarily being the theoretically optimal action (Sec. 6.1).
>
> c) We'll expand the discussion section with:
>
> > While SHML attempts to find optimal adaptations, it does not theoretically guarantee that the adaptations are indeed optimal. While we see this as a substantial improvement over reason-agnostic methods, future research could explore how to obtain theoretical guarantees of optimality within self-healing machine learning.
>
> ---
> # Thank you
>
> You have helped us improve our paper. Given these changes, we hope you consider revising your assessment of the paper's impact and the corresponding evaluation for NeurIPS. ☺️

---

> > ### Author Response · Authors · 2024-08-11
> >
> > Dear reviewer LnCp,
> >
> > We thank you once again for the effort put into reviewing our paper. As there are only a couple working days left in the discussion period, we would like to ask if our response has satisfied your concerns. If so, we hope you consider revising your assessment of the paper's impact and the corresponding evaluation for NeurIPS. If any concerns remain, we are happy to discuss them further here.

---

> > > ### Author Response · Authors · 2024-08-13
> > >
> > > Dear reviewer LnCp,
> > >
> > > Thank you for your thorough review of our work. As the discussion period is coming to a close, we'd like to follow up on your prior concerns.
> > >
> > > Have our responses and clarifications addressed your main questions and concerns? If so, we kindly ask that you consider increasing your score to better reflect this. If not, please let us know what specific issues remain, and we'll promptly provide additional information to resolve them.
> > >
> > > We appreciate your reconsideration and look forward to your feedback.

---

### Author Rebuttal · Authors · 2024-08-07

We thank the reviewers for their insightful and positive feedback!

We are encouraged by the unanimous recognition of our self-healing ML framework's importance and novelty. The reviewers consistently described our work as "important" (**R-LnCp**, **R-4TYN**, **R-xkmy**) with a "novel and interesting" approach (**R-LnCp**). Our paper had a "well-structured presentation" (**R-4TYN**) and was considered as "well organised and easy to read" (**R-vweT**), with the structure "logical and easy to follow" (**R-xkmy**). Reviewers mentioned that our paper is "outperforming existing approaches in both presentation and results" (**R-4TYN**). We would like to note the consensus on two key themes we saw across the reviews:

- **Innovative and impactful approach**. Reviewers noted our work "lowers the barriers for others to implement adaptation actions in machine learning models" (**R-4TYN**) and has "significant potential in high-stakes applications where maintaining optimal model performance is critical, such as healthcare and finance" (**R-xkmy**). "Rather than using reason-agnostic methods" (**R-xkmy**), we show that "the reasons for performance drop should be considered and investigated" (**R-vwet**). In this way, our paper addresses "the growing challenge of maintaining machine learning models, especially given their increasing usage" (**R-4TYN**).

- **Robust theoretical foundation**. The paper provides a "solid theoretical foundation" (**R-xkmy**) and the method "is described with precise formalism" (**R-vweT**). This is complemented by empirical evaluations demonstrating its practical viability (**R-xkmy**). Reviewers saw our evaluation as "strong and thorough, which covers the details to help readers further understand the scope and the logic of the proposed method" (**R-LnCp**). Reviewers appreciated the "detailed inclusion of examples, definitions, assumptions, theoretical explanations, and results in the main text and appendix" which are "thorough and effective" (**R-4TYN**).


---
# Information in the supplementary pdf
We provide in total five new viability studies. That said, we would like to note that we see the primary contribution of our paper as formalizing self-healing machine learning which researchers and practitioners can use to build their own self-healing systems.

We provide the following information in the supplementary pdf.
- **Table 1.** Additional viability studies involving **five real world datasets**. The datasets cover a wide variety of setups: Airlines (Bifet et al., 2010), Poker (Cattral et al. 2007), Weather (Elwell & Polikar, 2011), Electricity (Zliobaite, 2013), Forest Type (Blackard, 1998). We simulate real-world unexpected degradations by corrupting features at test time and evaluating models for different number of corrupted features and corruption values. **Takeaway**: $\mathcal{H}$-LLM better adapts at test time to issues that require reasoning about the structure of the data generating process across five datasets. This showcases the need for using self-healing systems in real-world environments.
- **Figure 1**. **Empirical insights** into the effect of self-healing on downstream accuracy for each of the **five datasets**. We systematically vary the corruption value and the number of corrupted columns and quantify the accuracy with and without triggering a self-healing mechanism. **Takeaway**: We find that the effect of applying a healing mechanism is largest when there is greatest corruption levels in the dataset. Furthermore, we find that applying a healing mechanism consistently improves downstream accuracy in the presence of data corruption. This showcases the importance of healing mechanisms in test-time environments.
- **Table 2**. Additional viability study involving **more benchmarks** on the original diabetes prediction task. We benchmark against an additional adaptation methods and four other adaptive algorithms. Results shown on the original diabetes setup described in the paper. We do not vary corruption values for space reasons. **Takeaway**: The additional benchmarks are unable to cope with adaptations when it requires reasoning about the structure of the data generating process.
- **Table 3**. **Qualitative ablation study** results for $\mathcal{H}$-LLM. We systematically remove one component of the system and inspect its outputs. **Takeaway**: All four components are required for self-healing to work. Removing any component results in poorer adaptations.
- **Table 4**. Additional viability study **evaluating each adaptation action across 10 different ML models**. We follow the same setup as in Table 1 by corrupting values at test time and varying the underlying model used. **Takeaway**: We show that self-healing is agnostic to the kind of model used. Self-healing ML can benefit any downstream ML model whenever understanding the reason for degradation is important (such as test time corruption).

---
# Thank you
The review has been extremely productive. We thank everyone for their help in shaping the paper to be in better form.

---

### Decision · Program_Chairs · 2024-09-25

**Decision:**

Accept (poster)

**Comment:**

This paper introduces a new approach to the problem of drift adaptation in ML models called Self-Healing Machine Learning (SHML). The strengths of the paper include: (1) high novelty in addressing an important problem - the SHML approach to drift adaptation starts from the assumption that drift adaption must be driven by the reasons (root causes) of the shift in the first place, in contrast to prior work on this problem; and (2) significant technical contributions - a solid theoretical framework is provided, along with compelling proof of principle experimental results to demonstrate the viability of the approach (particularly with the expanded comparative results that show the approach to outperform existing "reason-agnostic" approaches); The paper opens up several interesting possibilities for future research.

The main weaknesses of the paper relate to various problems with aspects of the presentation, including such issues as aspects of the approach being unclear, the fact that in other places the information needed to properly understand aspects of the work are buried in the supplementary material, that the paper could do a better job of motivating the approach and highlighting its novelty from prior work upfront, and that the writing could be made more concise in some places. The authors in their rebuttal propose changes that largely address these deficiencies and correct various reviewer misconceptions and concerns caused by these problems. Please include all of these changes in the final version of the paper, along with the expanded comparative performance results mentioned above.